# Molecular coordination of *Staphylococcus aureus* cell division

Victoria A Lund[1,2†], Katarzyna Wacnik[1,2†], Robert D Turner[1,2,3†], Bryony E Cotterell[1,2,4], Christa G Walther[1,2], Samuel J Fenn[1,2], Fabian Grein[5], Adam JM Wollman[6], Mark C Leake[6], Nicolas Olivier[1,3], Ashley Cadby[1,3], Stéphane Mesnage[1,2], Simon Jones[4], Simon J Foster[1,2]*

[1]Krebs Institute, University of Sheffield, Sheffield, United Kingdom; [2]Department of Molecular Biology and Biotechnology, University of Sheffield, Sheffield, United Kingdom; [3]Department of Physics and Astronomy, University of Sheffield, Sheffield, United Kingdom; [4]Department of Chemistry, University of Sheffield, Sheffield, United Kingdom; [5]Institute for Pharmaceutical Microbiology, German Center for Infection Research (DZIF), University of Bonn, Bonn, Germany; [6]Biological Physical Sciences Institute, University of York, York, United Kingdom

**Abstract** The bacterial cell wall is essential for viability, but despite its ability to withstand internal turgor must remain dynamic to permit growth and division. Peptidoglycan is the major cell wall structural polymer, whose synthesis requires multiple interacting components. The human pathogen *Staphylococcus aureus* is a prolate spheroid that divides in three orthogonal planes. Here, we have integrated cellular morphology during division with molecular level resolution imaging of peptidoglycan synthesis and the components responsible. Synthesis occurs across the developing septal surface in a diffuse pattern, a necessity of the observed septal geometry, that is matched by variegated division component distribution. Synthesis continues after septal annulus completion, where the core division component FtsZ remains. The novel molecular level information requires re-evaluation of the growth and division processes leading to a new conceptual model, whereby the cell cycle is expedited by a set of functionally connected but not regularly distributed components.
DOI: https://doi.org/10.7554/eLife.32057.001

**\*For correspondence:**
s.foster@sheffield.ac.uk

[†]These authors contributed equally to this work

**Competing interests:** The authors declare that no competing interests exist.

## Introduction

In order to grow and divide, bacteria must make new cell wall, the major structural component of which is peptidoglycan (*Turner et al., 2014*). Bacteria generally have two groups of proteins that co-ordinate peptidoglycan insertion, one involved with elongation (elongasome), the other with division (divisome) (*Cabeen and Jacobs-Wagner, 2005*). *S. aureus* lacks an apparent elongasome machinery, but nonetheless new peptidoglycan is inserted all over the cell surface, throughout the cell cycle, not just during cell division (*Monteiro et al., 2015*; *Zhou et al., 2015*). Addition of peptidoglycan, along with its hydrolysis (*Wheeler et al., 2015*), is what enables *S. aureus* cells to get bigger – volume increases at a constant rate (*Zhou et al., 2015*).

The *S. aureus* divisome contains both enzymes that catalyse addition of new monomers to the peptidoglycan envelope (penicillin-binding proteins, PBPs), and proteins that co-ordinate this activity. Chief amongst these is FtsZ - an essential protein in almost all bacteria that directs cell division, which has recently been shown to form dynamic filaments that 'treadmill' in *Escherichia coli* and *Bacillus subtilis*, giving a framework to assemble other division proteins resulting in cell wall biosynthesis and septum formation (*Yang et al., 2017*; *Bisson-Filho et al., 2017*). FtsZ assembly into the Z-ring is regulated by other cell division components including EzrA (*Levin et al., 1999*; *Adams and*

*Errington, 2009*), a membrane protein crucial for cell division in *S. aureus* (*Steele et al., 2011*). It has been shown to interact with both cytoplasmic proteins and those with periplasmic domains and it is therefore proposed to act as an interface between FtsZ and PBPs forming a scaffold for other cell division components (*Steele et al., 2011*).

Previously, FtsZ and EzrA in *S. aureus* have been imaged using fluorescent fusions (*Strauss et al., 2012*; *Pereira et al., 2016*) and sites of peptidoglycan insertion using fluorescent D-amino acids (*Monteiro et al., 2015*; *Kuru et al., 2012*). Here, we have applied single molecule localisation microscopy, a technique that provides unprecedented detail compared with other approaches. This has revealed an unexpected arrangement of division proteins and associated peptidoglycan insertion pattern. This defies the conventional view of division in *S. aureus* and has prompted a new model that encompasses the morphological idiosyncrasies of this important pathogen.

## Results

### Distribution of divisome components during septation

In order to visualise division machines, we localised the cytoplasmic initiator of division FtsZ and the crucial membrane protein EzrA (*Steele et al., 2011*). Four fusions of EzrA with different fluorophores were created. These had wild-type growth rates and the previously observed septal EzrA localization pattern (*Steele et al., 2011*; *Jorge et al., 2011*) by diffraction limited microscopy (*Figure 1—figure supplement 1*). Localisation microscopy and 3D structured illumination microscopy (3D-SIM) were used to address the distribution and juxtaposition of the cell division components at super-resolution.

3D-SIM revealed that EzrA exhibited punctate distribution at the division site (*Figure 1—figure supplement 2a*) (*Strauss et al., 2012*). Unfortunately, the 'honeycomb' artefact (which introduces foci in images due to incomplete noise filtering [*Komis et al., 2015*]), visible in our images, could not be removed by raising the Weiner filter parameter in reconstructions. Thus, localisation microscopy was employed as a superior approach.

eYFP was selected as a blinking fluorescent protein tag (*Biteen et al., 2008*). Multiple 2D images of septa in the plane of focus were obtained for EzrA-eYFP (*Figure 1a*), FtsZ-eYFP (*Figure 1b*) and EzrA-meYFP (*Figure 1—figure supplement 2b*). The mean localisation precision of eYFP was calculated using two different formulae: the 'Thompson Equation' (*Thompson et al., 2002*) by the ThunderSTORM ImageJ plugin yielded 24 (s.d. 8.5) nm while a using a modified version of this equation (*Mortensen et al., 2010*) yielded 27 (s.d. 8.7) nm. We also measured it experimentally using Nearest Neighbour in Adjacent Frames (NeNA) analysis (*Endesfelder et al., 2014*): NeNA analysis determines localisation precision based on spatial proximity of blinks that occur at similar times and is part of a family of clustering-based tools for assessing the quality of localisation microscopy data (*Coltharp et al., 2012*). This method gave us a mean localization precision of 16.2 nm. Many of the septa appeared to be somewhat elliptical. This is likely due to the cells being tilted relative to the plane of focus leading to circular septa appearing elliptical. We therefore fitted ellipses to the septal localisations and calculated the expected tilt of the cells. The results were that all of the localisations included in our analysis are within a 400 nm optical section, within a range to ensure good data (*Palayret et al., 2015*).

To analyse the distributions and address issues of sampling and resolution in our microscopy, a number of simple simulations were carried out where representative numbers of localisations were distributed at random in rings of similar radius to those observed, with a random error applied (*Figure 1c*). A circle was fitted to the data points and all the distributions (experimental and simulated) were parameterised with respect to angle and distance from the centre of the circle, generating histograms of localisations (*Figure 1d,e*). The autocorrelations of the angular distributions were then averaged to show that the localisations in the experimental data were neither completely randomly, or regularly, distributed around the ring (*Figure 1f*). Distributions of distance from the centre of the circle were compared with simulated distributions of a fixed circle radius where different levels of localisation precision error were applied (*Figure 1g*). Even with the most conservative assumptions (including simulated localisation precisions worse than we had calculated for our measured data), the localisations were spread out over a sufficiently wide range of distances to indicate that both FtsZ or EzrA do not form a very thin ring at the leading edge of the septum in *S. aureus*.

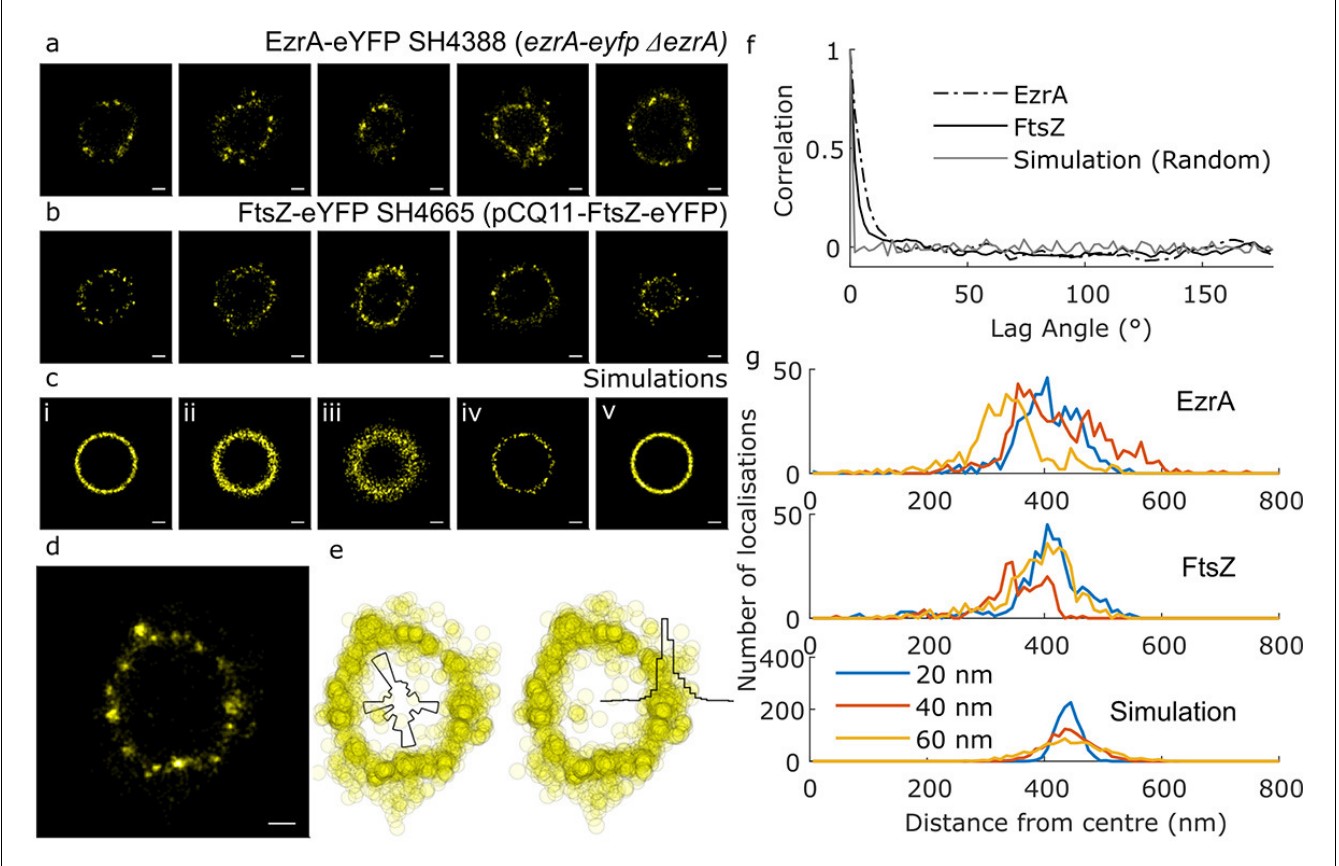

**Figure 1.** Distribution of cell division components during septation. (a) Examples of EzrA distributions obtained using localisation microscopy of SH4388 (*ezrA-eyfp ΔezrA*). Scale bars 200 nm. (b) Examples of FtsZ distributions obtained using localisation microscopy of SH4665 (pCQ11-FtsZ-eYFP) grown with 50 μM IPTG. Scale bars 200 nm. (c) Simulated distributions of localisations randomly distributed by angle with different radii (r), number of localisations (n) and random error from a normal distribution with standard deviation (σ) [i] r = 440 nm, n = 1118, σ = 20 nm, [ii] r = 440 nm, n = 1118, σ = 40 nm, [iii] r = 440 nm, n = 1118, σ = 80 nm, [iv] r = 440 nm, n = 145, σ = 20 nm, [v] r = 440 nm, n = 2010, σ = 20 nm. Scale bars 200 nm. (d) An enlarged example of EzrA-eYFP distribution. Scale bar 200 nm. (e) The distribution from 'd' plotted as a scatter graph, and as histograms of number of localisations with respect to angle and distance from centre. (f) Mean angular autocorrelations of 14 EzrA, 19 FtsZ and 15 simulated distributions. Autocorrelation drops less quickly for EzrA and FtsZ than for simulations where angle is randomised. This shows that neither EzrA or FtsZ are randomly distributed by angle. (g) Histograms of localisations with respect to distance from the centre of a fitted circle with varying localisation precision. Data for EzrA and FtsZ are spread more widely than simulated data with poor localisation precision.

DOI: https://doi.org/10.7554/eLife.32057.002

The following figure supplements are available for figure 1:

**Figure supplement 1.** EzrA fusions are functional.

DOI: https://doi.org/10.7554/eLife.32057.003

**Figure supplement 2.** STORM and SIM data.

DOI: https://doi.org/10.7554/eLife.32057.004

**Figure supplement 3.** Quantitative analysis of EzrA and FtsZ distributions from localisation microscopy data based on elliptical fits.

DOI: https://doi.org/10.7554/eLife.32057.005

**Figure supplement 4.** Dynamics of EzrA.

DOI: https://doi.org/10.7554/eLife.32057.006

Instead both proteins appear in a non-uniform distribution within the septal annulus. Within the annulus the proteins show no discernible pattern within or across cells. FtsZ distributions were consistent with FtsZ remaining in the division plane after septal fusion were also observed (*Figure 2a*).

To further investigate whether the apparent elliptical shape of the rings had an influence on our interpretation, we also analysed the data using an elliptical, rather than a circular fit. Comparing our results to simulated data (*Figure 1—figure supplement 3*) corroborated our previous findings.

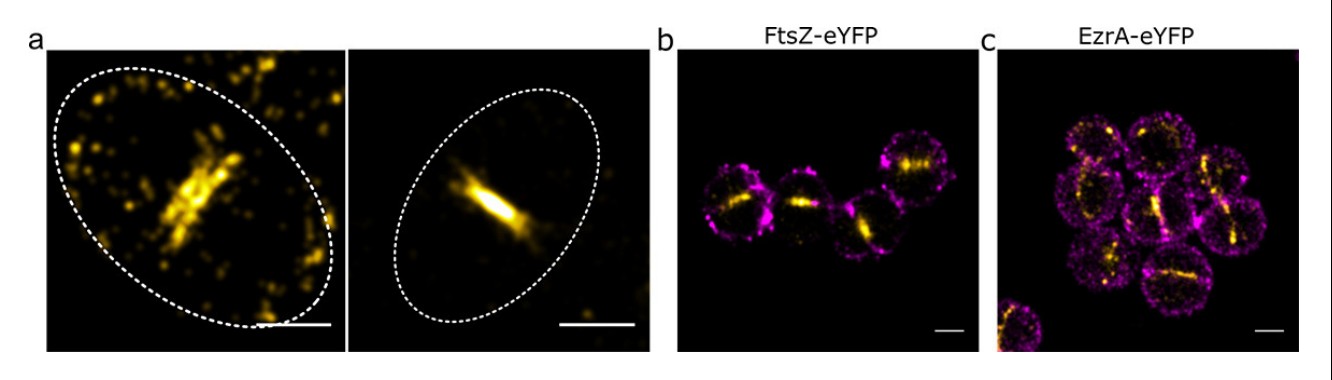

**Figure 2.** Relative locations of division components. (**a**) Localisation microscopy images: of FtsZ-eYFP distributions in bacteria in the late stages of division. Scale bars 500 nm. Ellipses show approximate cell location and orientation. (**b**) Dual colour localisation microscopy image of FtsZ-eYFP and the cell wall (labelled with Alexa Fluor 647 NHS ester, NHS-647). Scale bars 500 nm. (**c**) Dual colour localisation microscopy image of EzrA-eYFP and the cell wall (labelled with NHS-647). Scale bars 500 nm.

DOI: https://doi.org/10.7554/eLife.32057.007

To place these findings in the context of cell wall shape, two colour localisation microscopy was performed where the cell wall was labelled with an Alexa Fluor 647 NHS ester (*Figure 2b,c*), which labels all amine groups in the cell wall (*Zhou et al., 2015*). This confirmed that EzrA and FtsZ were at the expected septal positions in the cell.

To analyse rapid molecular dynamics of EzrA, single-molecule Slimfield microscopy (*Plank et al., 2009*) was performed on EzrA-meYFP labelled *S. aureus*, SH4604 (*ezrA-meyfp ΔezrA*) optimized to enable blur-free tracking of single fluorescent protein fusion constructs in live cells over a millisecond timescale (*Reyes-Lamothe et al., 2010*; *Badrinarayanan et al., 2012*). Analysis of the mobility of tracked EzrA-meYFP foci enabled quantification of their microdiffusion coefficient (D), indicating a mixture of three different mobility components: an apparent immobile population in addition to an intermediate and a rapid mobility population (*Figure 1—figure supplement 4a,b*). In total, ~600 EzrA foci tracks were analysed in the septum region, whose overall mean D value, which captures both the immobile and two mobile populations, was $0.20 \pm 0.01\ \mu m^2\ s^{-1}$. Whereas, 140 foci tracks were detected outside the septum region, which showed an increased overall mean D of $0.28 \pm 0.03$ $\mu m^2\ s^{-1}$. This greater average mobility was principally due to an increase in the proportion of EzrA foci present in the most mobile component (going from $33 \pm 3\%$ of the total to $42 \pm 4\%$).

These relatively slow mobility values for EzrA, compared to many freely diffusing bacterial membrane integrated proteins (*Leake et al., 2008*), do not preclude putative rotational/treadmilling motions of EzrA (which have been observed in previous studies of FtsZ mobility in *E. coli* and *B. subtilis* [*Yang et al., 2017*; *Bisson-Filho et al., 2017*]) over a longer time scale. For example, the mean speed of putative FtsZ treadmilling estimated from *B. subtilis* recently (*Bisson-Filho et al., 2017*) is only ~30 nm/s, which we estimate would be sufficiently slow to appear predominantly in the immobile component over the typical time scales of our Slimfield tracking experiments here, and so putative treadmilling of EzrA at this equivalent mean speed, if present in *S. aureus*, would most likely appear in this apparent immobile fraction. However, in the three component mobility model, which fits the observed distribution of D values well, the intermediate mobility fraction has been interpreted previously in other cellular systems as indicating transient dynamic interactions (*Stracy et al., 2015*), and so we cannot entirely exclude the possibility that this may be due to transient association of EzrA with FtsZ. Deconvolution analysis (*Wollman and Leake, 2015*) of whole cell images obtained using Slimfield microscopy indicated a mean total copy number of $305 \pm 23$ EzrA molecules per cell measured across a population (Figure 1—figure supplement 4c). Estimating the proportion of the most mobile fraction of EzrA foci therefore indicates that at least ~100 EzrA molecules per cell are not likely to be treadmilling in tight association with FtsZ. In other words, we cannot account for the observed mobility of EzrA by a simple treadmilling model alone in which all EzrA is tightly associated with FtsZ, rather the real cellular behaviour is more complex than this.

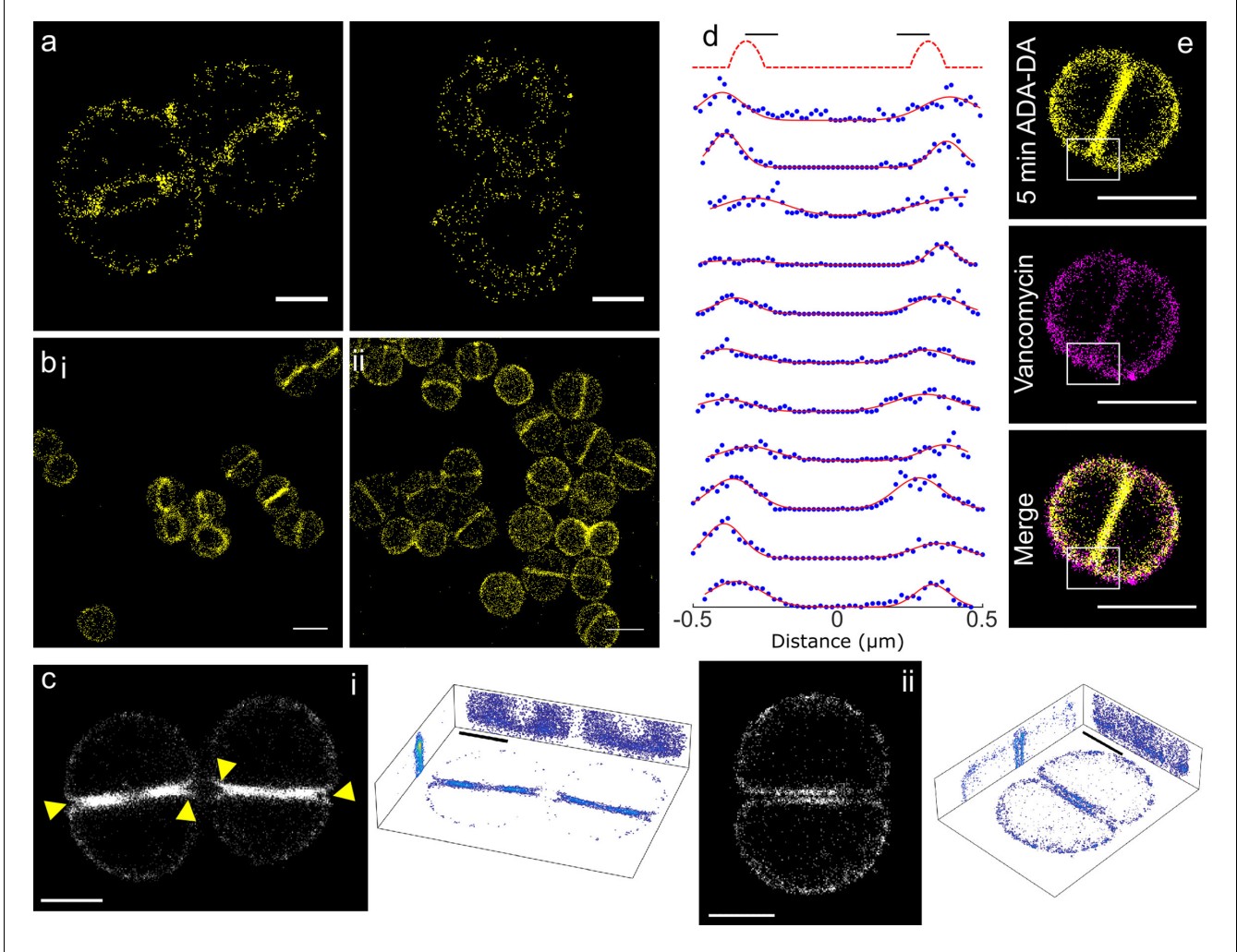

**Figure 3.** Peptidoglycan insertion. Localisation microscopy images: (**a**) 15 s labelling of ADA clicked to Alexa Fluor 647. Scale bars 0.5 μm. (**b**) 5 min labelling of (i) ADA clicked to Alexa Fluor 647 and (ii) ADA-DA clicked to Alexa Fluor 647. Scale bars 1 μm. (**c**) 3D projections of *S. aureus* labelled for 5 min with ADA clicked to Alexa Fluor 647. (i) Cells with incomplete septum (yellow arrows show gaps in labelling), (ii) cell with annulus complete. Images in black boxes are z-projections while 3D representations show projections in all three planes. Scale bar 0.5 μm. (**d**) Cross sections of incomplete septa. The sketch graph (top row) hypothetically shows labelling exclusively at the leading edge of the septum. This is not the case for the data shown below - labelling is spread throughout the septum. The full width half maximum spread of labelling is ~230 nm. Data are plotted with blue dots, fits in red lines. (**e**) Two colour STORM, sample labelled for 5 min with ADA-DA clicked to Alexa Fluor 647 (yellow) and vancomycin linked to Amersham Cy3B (magenta). Images are z-projections and in merged images where localisations are in white show labelling by both ADA-DA and vancomycin. Boxed regions show slot in ADA-DA labelling but not vancomycin. Scale bars 1 μm.

DOI: https://doi.org/10.7554/eLife.32057.008

The following figure supplements are available for figure 3:

**Figure supplement 1.** Identification of mechanism of DAA labelling in *S. aureus*.
DOI: https://doi.org/10.7554/eLife.32057.009

**Figure supplement 2.** 15 s labelling of peptidoglycan insertion with DAAs and controls.
DOI: https://doi.org/10.7554/eLife.32057.010

**Figure supplement 3.** DAA labelling of PBP4 null *S. aureus*.
DOI: https://doi.org/10.7554/eLife.32057.011

## Peptidoglycan synthesis in *S. aureus* does not occur in discrete foci

We used established metabolic labelling with fluorescent D-amino acids or dipeptides (*Monteiro et al., 2015*; *Kuru et al., 2012*) and adapted this for localisation microscopy to visualise peptidoglycan insertion with this higher resolution imaging technique. We confirmed that HADA (7-

hydroxycoumarin-3-carboxylic acid-amino-D-alanine), ADA (azido D-alanine) and ADA-DA (azido-D-alanyl-D-alanine) mark regions of new peptidoglycan insertion by microscopy and liquid chromatography-mass spectrometry (LC-MS) (*Figure 3—figure supplement 1*).

Cells were pulse labelled with DAAs (D-amino acids) from 15 s to 5 min. Even at the very shortest labelling time (15 s) peptidoglycan synthesis was observed both at the septum and cell periphery but without discrete foci (*Figure 3—figure supplement 2a,b*). Localisation microscopy of 15 s ADA and ADA-DA labelled cells revealed labelling occurs dispersed across the whole septum as well as the off-septal cell wall (*Figure 3a*, *Figure 3—figure supplement 2d*). This was not due to non-specific labelling (*Figure 3—figure supplement 2c*). XY localisation precision (estimated by the Nikon N-STORM software) was 9.9 (s.d. 3.5) nm or 7.5 nm by NeNA (*Endesfelder et al., 2014*). A similar pattern of peptidoglycan synthesis was seen with up to 5 min labelling with ADA or ADA-DA as a zone across the developing septum as well as throughout the off-septal cell wall (*Figure 3b,c,d*). Previously PBP4 has been implicated in the presence of off-septal incorporation (*Monteiro et al., 2015*; *Gautam et al., 2015*), we therefore carried out DAA labelling and localisation microscopy in a PBP4 null background (SH4425) (*Figure 3—figure supplement 3*). Cell growth and GlcNAc incorporation were found to be the same as WT, however DAA labelling was reduced in SH4425 (*Figure 3—figure supplement 3b–d*). The proportion of off-septal labelling was calculated in both SH1000 and SH4425 when labelled with ADA-DA, however no significant difference was observed (*Figure 3—figure supplement 3e*). Localisation microscopy of both 15 s and 5 min labelled SH4425 showed peptidoglycan synthesis both at the septal and peripheral cell wall. Discrete foci of insertion were not observed (*Figure 3—figure supplement 3f–g*). Comparison of autocorrelations (as calculated for EzrA and FtsZ, using elliptical fits) for SH1000 and SH4425 revealed no substantial differences (*Figure 3—figure supplement 3h*).

In cells with an incomplete septum, there was a 'gap' in peptidoglycan synthesis at the mother cell wall-septum interface (*Figure 3c–i*, arrows). In order to investigate the properties of the observed 'gap' we used a counter stain to determine if it is filled with peptidoglycan. Fluorescent vancomycin has been used extensively to label peptidoglycan (*Daniel and Errington, 2003*). Thus, we synthesised a version of this molecule with a Cy3B fluorophore so it could be used in two colour localisation microscopy with Alexa Fluor 647 click tagged amino acids. Vancomycin binds D-alanyl-D-alanine motifs in peptidoglycan and as these are highly prevalent in *S. aureus* the majority of peptidoglycan is fluorescently labelled. Our two colour images show that the 'gap' regions that do not contain ADA-DA (5 min labelling), are nonetheless bound by vancomycin and thus are filled with peptidoglycan (*Figure 3e*).

Also, cells with a filled septal annulus showed continued insertion that could be resolved into two distinct zones, one for each daughter (*Figure 3c–ii*). These features were not observable by SIM, being smaller than its theoretical resolution.

## Inhibition of cell division leads to co-mislocalization of the cell division components and peptidoglycan synthesis

The FtsZ inhibitor PC190723 prevents depolymerisation of FtsZ and consequently inhibits cell division, also leading to swollen *S. aureus* cells (*Haydon et al., 2008*). It has previously been shown by diffraction limited fluorescence microscopy that PC190723 causes mislocalisations of FtsZ and PBP2 (*Tan et al., 2012*). We sought to determine the dynamics of this process, and the molecular pattern of associated peptidoglycan insertion. PC190723 treatment led to delocalization of peptidoglycan biosynthesis, EzrA and FtsZ even before substantial cell swelling (*Figure 4—figure supplement 1*). Incorporation of HADA does not cause mislocalisation of FtsZ or EzrA (data not shown). Peptidoglycan synthesis was observed around the cell periphery and in distinct foci in the same place as EzrA and FtsZ. This non-uniform peptidoglycan insertion results in misshapen cells with irregular thickening of the cell wall (*Figure 4a*). After 60 min treatment, patches of FtsZ, EzrA and peptidoglycan synthesis can be seen (*Figure 4—figure supplement 1a*). Localisation microscopy of peptidoglycan synthesis shows cell shape and the off-septal synthesis with patches of increased synthesis more clearly (*Figure 4b*). Thus, peptidoglycan synthesis follows localization of FtsZ and EzrA.

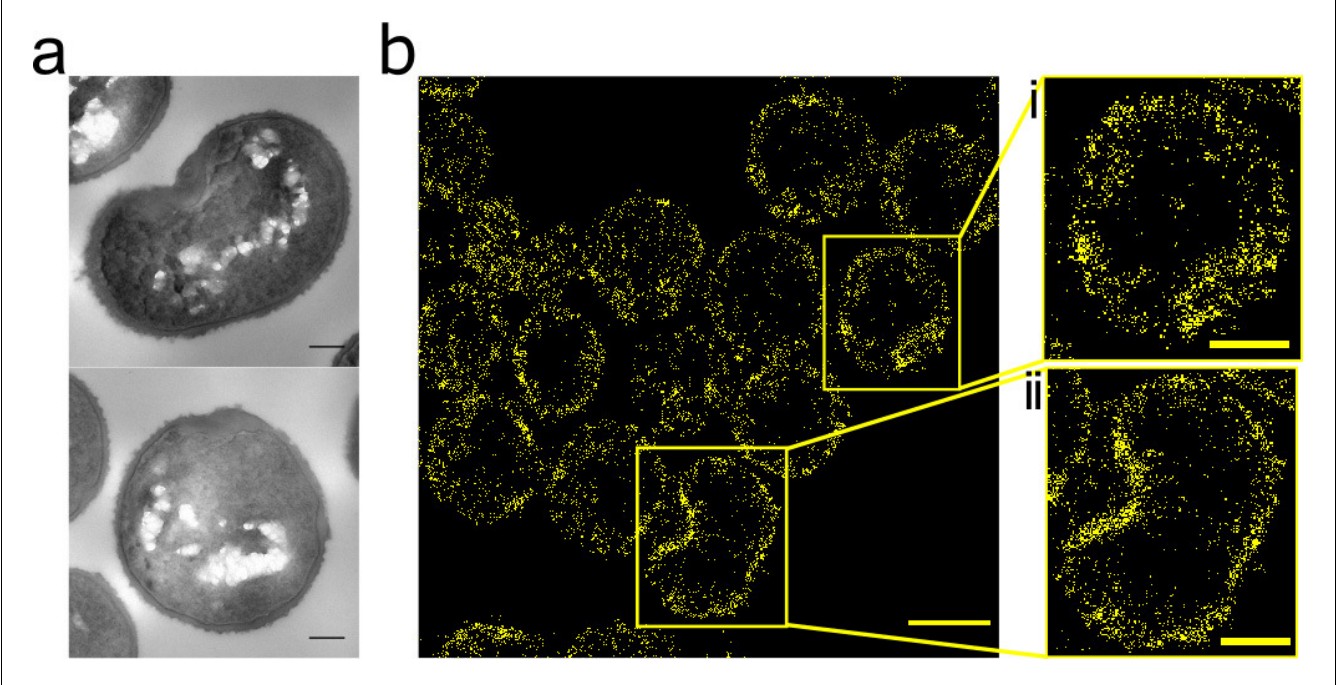

**Figure 4.** Effect of FtsZ inhibitor PC190723 on *S. aureus*. (a) TEM of *S. aureus* SH1000 grown in the presence of PC190723 (10 μg ml$^{-1}$) for 60 min. Scale bars 200 nm. (b) STORM image of *S. aureus* SH1000 pre-treated with PC190723 (10 μg ml$^{-1}$) for 60 min labelled with ADA clicked to Alexa Fluor 647 for 5 min. Scale bar 1 μm. (i) and (ii) zoomed images of the corresponding area, scale bars 0.25 μm.

DOI: https://doi.org/10.7554/eLife.32057.012

The following figure supplement is available for figure 4:

**Figure supplement 1.** Effect of FtsZ inhibitor PC190723 on *S. aureus*.

DOI: https://doi.org/10.7554/eLife.32057.013

## Morphology of the *Staphylococcus aureus* septum

It has been shown that the incomplete *S. aureus* septum is thinner at the leading than at the lagging edge (*Giesbrecht et al., 1998*; *Matias and Beveridge, 2007*). However, the significance of this has remained unknown. We observed sections of cells from different stages in the cell cycle and measured septal geometry using thin section Transmission Electron Microscopy (TEM). The septum of *S. aureus* is thinner at the leading edge and progressively thicker towards the lagging edge until it fuses, at which point it is thinner at the centre and progressively thicker towards the lagging edge until ultimately uniform thickness is established (*Figure 5a,b*). This dictates that peptidoglycan insertion cannot be confined to the leading edge of the septum and gives a morphological explanation for the observed peptidoglycan insertion pattern.

The surface area available for peptidoglycan insertion in the nascent septum was modelled resulting in the following expression for septal surface area prior to fusion (*Figure 5c*):

$$A = \pi(2r - s)\sqrt{s^2 + d^2}$$

Where $d$ is half the thickness of the septum, $r$ is the radius of the cell in the plane of septation and $s$ is the distance from the leading to the lagging edge of the septum (measured from the inner surface of the cell wall).

The surface area of a septum with consistently uniform thickness is that of the leading edge of that septum:

$$A = 4\pi(r - s)d$$

Not only is the available surface area always larger for the morphology we observe, but it increases as the septum closes (whereas with a uniformly thick septum, it decreases). This provides a

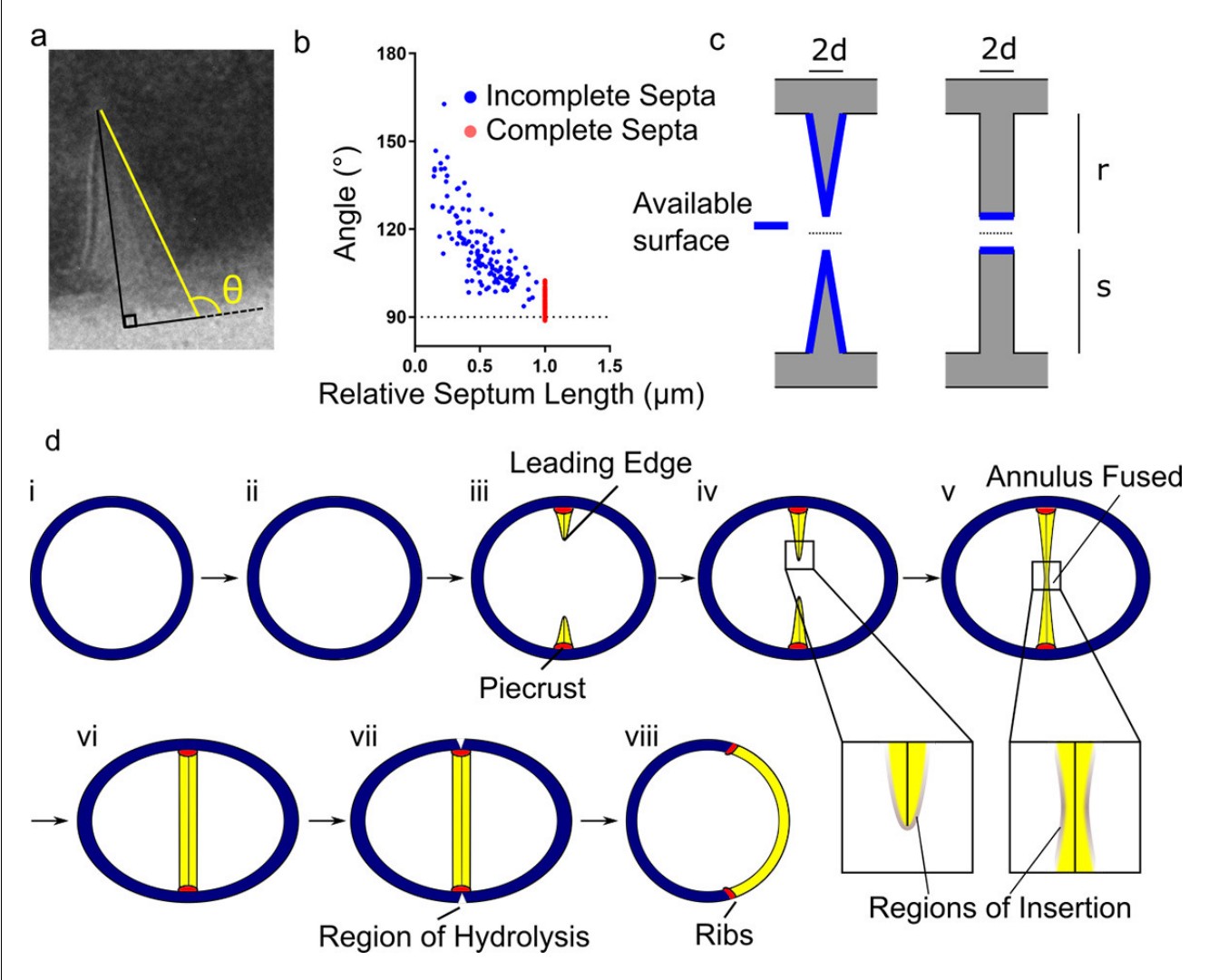

**Figure 5.** Conceptual model of peptidoglycan insertion during the *S. aureus* cell cycle. (**a**) Schematic of measurement used in (**b**) measurement of the angle (θ) between a line parallel to the surface of the septum (yellow) and a tangent to the surface of the bacterium in incomplete (blue) and complete (red) septa. (**c**) Surfaces available for peptidoglycan insertion for different septal geometries where d is half the thickness of the septum, r is the cell radius in the septal plane and s is the distance from the leading to the lagging edge of the septum (measured from the inner surface of the cell wall). (**d**) Conceptual model of peptidoglycan insertion in *S. aureus*. (i, ii) Cell size increases and aspect ratio changes prior to observation of the start of septum formation by 3D-SIM (***Monteiro et al., 2015***). (iii) The septum then starts to form, beginning with the 'piecrust' feature (red) observed by AFM (***Turner et al., 2010***). The septum is thinner at the leading edge (***Matias and Beveridge, 2007***). (iv) New peptidoglycan is inserted in a zone at the leading edge of the septum, as well as across the rest of the cell surface as visualised here by localisation microscopy. (v, vi) After the annulus has fused, peptidoglycan insertion continues in the septum, executed by cell division components, until it is of uniform thickness. (vii) ATL (a peptidoglycan hydrolase) is present at the outer surface of the cell in the plane of septation(***Komatsuzawa et al., 1997***). Cracks or splits begin to form at the outer surface in the plane of septation(***Touhami et al., 2004***), followed by rapid popping apart of the daughter cells (***Zhou et al., 2015***). (vii) 'Scars' or 'ribs' remain marking the site of division (***Monteiro et al., 2015***; ***Turner et al., 2010***) and may provide spatial cues to subsequently enable correct sequentially orthogonal divisions.

DOI: https://doi.org/10.7554/eLife.32057.014

framework for septal synthesis in an organism in which the septum comprises a substantial proportion of the cell wall.

## Discussion

The non-standard cross section of the septum in *S. aureus* distinguishes it from other model organisms (*Figure 5a,b,c*) and indicates that not all peptidoglycan insertion occurs at the leading edge of the septum in this species prompting the development of a new model for how peptidoglycan is synthesised during the cell cycle (*Figure 5d*). This is likely advantageous to the bacteria, enabling more biosynthetic enzymes to work on the cell wall without steric hindrance. We sought to explain this phenomenon by analysing the distribution of peptidoglycan insertion and investigating key cell division components. Our novel application of localisation microscopy to DAAs revealed that even at the shortest timescales and with considerably more precision than previous studies (*Monteiro et al., 2015*; *Zhou et al., 2015*; *Turner et al., 2010*), there were no foci of peptidoglycan insertion – the diffuse pattern throughout the septum and periphery of the cell was ever-present. This surprising finding was corroborated by the distribution of core cell division components in *S. aureus*. Localisation microscopy of FtsZ and EzrA in the septal ring showed, like the distribution of peptidoglycan insertion, that they occurred in a zone, and were not limited to the leading edge of the septum. Also, FtsZ remained at the septum after the annulus had fused. When FtsZ depolymerisation was inhibited, peptidoglycan insertion was found to occur in areas with large amounts of FtsZ, resulting in local thickening of the cell wall, suggesting all synthesis may depend on FtsZ. This is a different scenario to *E. coli* and *B. subtilis*, where division-associated foci of peptidoglycan synthesis have been identified (albeit without the precision of localisation microscopy) and associated with cell division components driven by treadmilling FtsZ filaments (*Yang et al., 2017*; *Bisson-Filho et al., 2017*).

The divisome has been proposed to be a multi-component machine, present within a ring, based on diffraction-limited microscopy and interaction studies (*Steele et al., 2011*; *Bottomley et al., 2014*). Previous localisation microscopy studies have begun to reveal intricate structural and spatial relationships between division components (*Holden et al., 2014*; *Buss et al., 2015*; *Jacq et al., 2015*).

Our data show that divisome components are not placed exclusively at the leading edge of the septum, and that some individual proteins move more rapidly than others. There may, therefore, be a number of essentially identical machines executing peptidoglycan insertion within a region of the septum, with exchange of machine components with a more mobile population of molecules. It could also be the case that the machines are very non-uniform and can execute their tasks with a subset of the complete list of divisome proteins and with more or less of an individual protein. Alternatively, stable, stoichiometric complexes are not present and the interactions between proteins required to make new peptidoglycan are highly transient.

## Materials and methods

### Bacterial growth conditions

Strains used in this study are listed in *Appendix 1—table 1*, while plasmids and oligonucleotide sequences are shown in *Appendix 1—table 2* and *Appendix 1—table 3*. *S. aureus* was grown in Brain Heart Infusion (BHI) broth at 37°C with aeration at 250 rpm, except for Slimfield microscopy and $^{14}$C-GlcNAc incorporation experiments (and associated growth curves) which were carried out using Chemically Defined Media (CDM) (*Hussain et al., 1991*). For solid media 1.5% (w/v) agar was added. Where required, antibiotics were added at the following concentrations; erythromycin (5 µg ml$^{-1}$), lincomycin (25 µg ml$^{-1}$), kanamycin (50 µg ml$^{-1}$), and tetracycline (5 µg ml$^{-1}$). To induce protein production strains carrying gene fusions under the control of the Pspac promoter were grown in the presence of 50 µM isopropyl β-D-thiogalactopyranoside (IPTG).

### Construction of *S. aureus* mutants

All vectors were constructed in *E. coli* NEB5α (New England Biolabs) following previously described methods (*Sambrook and Russell, 2001*; *Gibson et al., 2009*). The resulting constructs were passed through a restriction-deficient *S. aureus* RN4220 before being transduced into a final *S. aureus* SH1000 strain. Transformation and phage transduction of *S. aureus* were carried out as described previously (*Schenk and Laddaga, 1992*; *Novick and Morse, 1967*).

**SH4388** (*ezrA-eyfp ΔezrA*): The EzrA-eYFP fusion was created by AscI and NotI digestion of pGM074 and insertion of *eyfp* amplified by PCR from SU492 (*Monahan et al., 2014*) using primer pair eYFP-F and eYFP-R. pGM074 is pKASBAR-kan (*Bottomley et al., 2014*) containing *ezrA* under the control of its own promoter with the C-terminal *psmorange* (flanked by AscI and NotI restriction sites). In the resulting plasmid pKASBAR-EzrA-eYFP the translational fusion of *ezrA-eyfp* is linked by linker A (see below). pKASBAR-EzrA-eYFP was electroporated into CYL316 (*Lee et al., 1991*) and its integration at the *geh* locus was confirmed by disruption of lipase production on Baird-Parker medium. The chromosomal fragment containing the integrated plasmid was moved into *S. aureus* SH1000 by phage transduction, creating SH4384 (*ezrA-eyfp*).

To delete native *ezrA*, an *ezrA* deletion vector was constructed. Fragments encompassing ~1.5 kb regions flanking *ezrA* were PCR amplified from *S. aureus* SH1000 genomic DNA using pOB-ezrA-up-F/-R and pOB-ezrA-down-F/-R. A 2.1 kb fragment encoding a tetracycline resistance cassette (*tetR*) was amplified from pAISH by PCR using pOB-TetR-F/-R primers. The three PCR products were ligated with HindIII and EcoRI cut pOB (*Horsburgh et al., 2002a*) by Gibson assembly, creating a deletion vector pOB-ΔezrA. The plasmid pOB-ΔezrA was electroporated into RN4220. The plasmid integrated into the chromosome through a single cross-over event and the DNA fragment containing the deletion cassette was transduced into SH4386 (*ezrA-eyfp*). Tetracycline-resistant/erythromycin-sensitive colonies were selected. In the resulting strain, SH4388 (*ezrA-eyfp ΔezrA*), *ezrA-eyfp* was the only copy of the *ezrA* gene. Replacement of *ezrA* for *tetR* was confirmed by PCR and Southern blot.

**SH4640** (*ezrA-gfp ΔezrA*): To construct an EzrA-GFP translational fusion linked by linker A, *gfp* was PCR amplified from JGL227 (*Steele et al., 2011*) using GFP-F/-R primers and ligated into AscI and EcoRI cut pGM074, creating pKASBAR-EzrA-GFP. The resulting plasmid was electroporated into CYL316. pKASBAR-EzrA-GFP integration at the *geh* locus was confirmed by disruption of lipase production on Baird-Parker medium. The chromosomal region containing the plasmid integrated within *geh* was moved to SH1000 creating SH4639 (*ezrA-gfp*). To delete native *ezrA*, SH4639 was transduced with a phage lysate from SH4388 (*ezrA-eyfp ΔezrA*), creating SH4640 (*ezrA-gfp ΔezrA*). Replacement of *ezrA* for TetR was confirmed by PCR and Southern blot.

**SH4642** (*ezrA-snap ΔezrA*): A translational fusion of EzrA linked by linker A to the SNAP tag was constructed by PCR amplification of *snap* from pSNAP-tag (T7)−2 (New England Biolabs) using SNAP-F/-R primers. The PCR product was ligated into pGM074 using AscI and NotI cut sites to create pKASBAR-EzrA-SNAP. The resulting plasmid was electroporated into CYL316, and its integration at the *geh* locus was confirmed by disruption of lipase production on Baird-Parker medium. The chromosomal fragment containing integrated pKASBAR-EzrA-SNAP was transduced into SH1000, resulting in SH4641 (*ezrA-snap*). Native *ezrA* was replaced by *tetR* by transducing SH4641 with the phage lysate from SH4388 (*ezrA-eyfp ΔezrA*), creating SH4642 (*ezrA-snap ΔezrA*). Replacement of *ezrA* for *tetR* was confirmed by PCR and Southern blot.

**SH4604** (*ezrA-meyfp ΔezrA*): To create a C-terminal fusion of EzrA with monomeric eYFP (meYFP) the whole pKASBAR-EzrA-eYFP plasmid was PCR amplified using meYFP-F/-R primers. The meYFP-F primer introduced an A206K substitution (*Zacharias et al., 2002*) into the *eyfp* gene. The PCR product was digested with DpnI to remove methylated DNA, the 5' ends of DNA were phosphorylated with T4 polynucleotide kinase (New England Biolabs) and DNA was circularized using Quick-Stick ligase (Bioline), resulting in pKASBAR-EzrA-meYFP. The resulting plasmid was electroporated into CYL316. The chromosomal fragment containing the integrated plasmid in the *geh* locus was moved into *S. aureus* SH1000 by phage transduction, creating SH4603 (*ezrA-meyfp*), To delete native *ezrA*, SH4603 was transduced with a phage lysate from SH4388 (*ezrA-eyfp ΔezrA*), creating SH4604 (*ezrA-meyfp ΔezrA*). Replacement of *ezrA* for *tetR* was confirmed by PCR and Southern blot.

**SH4652** (*ezrA-eyfp ΔezrA pCQ11-FtsZ-SNAP*): In order to construct a strain simultaneously producing EzrA-eYFP and FtsZ-SNAP, a plasmid encoding a translational *ftsZ-snap* fusion placed under the control of the Pspac promoter was constructed. The *ftsZ* gene was PCR amplified from *S. aureus* N315 genomic DNA using FGFtsZXhoI-F and FGFtsZEcoRI-R primers and cloned into EcoRI and XhoI cut pSS26b (Covalys), resulting in pSS26bFtsZ-C. The fragment encoding *ftsZ-snap* was PCR amplified from pSS26bFtsZ-C using FGFtsZNheI-F and FGFtsZAscI-R and inserted into pCQ11 (*Hardt et al., 2017*) using NheI and AscI cut sites, creating pCQ11-FtsZ-SNAP. The plasmid was electroporated into RN4220 and moved to SH4388 (*ezrA-eyfp ΔezrA*) by phage transduction, resulting in SH4652 (*ezrA-eyfp ΔezrA pCQ11-FtsZ-SNAP*).

**SH4665** (pCQ11-FtsZ-eYFP): To construct a translational fusion of FtsZ with eYFP, an insert containing a fragment of linker B (see below) followed by a full length *eyfp* gene was synthesized by the GeneArt Gene Synthesis service, PCR amplified using ftsZ-eyfp-F/-R primers and cloned into NcoI and AscI cut pCQ11-FtsZ-SNAP, creating pCQ11-FtsZ-eYFP. The plasmid was electroporated to RN4220 and moved to SH1000 by phage transduction, resulting in SH4665 (pCQ11-FtsZ-eYFP).

**SH4425** (*pbp4*): NE679 (*pbp4*) containing a transposon insertion within the *pbp4* gene was obtained from NARSA library (*Fey et al., 2013*). SH1000 was transduced with a phage lysate from NE679. Insertion of the transposon within *pbp4* in resulting SH4425 (*pbp4*) was confirmed by PCR and sequencing.

## Sequences of genes encoding fluorescent proteins, tags and linkers

*eyfp* in pKASBAR-EzrA-eYFP
ATGGTGAGCAAGGGCGAGGAGCTGTTCACCGGGGTGGTGCCCATCCTGGTCGAGC
TGGACGGCGACGTAAACGGCCACAAGTTCAGCGTGTCCGGCGAGGGCGAGGGCGATGCCACC
TACGGCAAGCTGACCCTGAAGTTCATCTGCACCACCGGCAAGCTGCCCGTGCCCTGGCCCACCC
TCGTGACCACCTTCGGCTACGGCCTGCAGTGCTTCGCCCGCTACCCCGACCACATGAAGCAG-
CACGACTTCTTCAAGTCCGCCATGCCCGAAGGCTACGTCCAGGAGCGCACCATCTTCTTCAAG-
GACGACGGCAACTACAAGACCCGCGCCGAGGTGAAGTTCGAGGGCGACACCCTGG
TGAACCGCATCGAGCTGAAGGGCATCGACTTCAAGGAGGACGGCAACATCCTGGGGCACAAGC
TGGAGTACAACTACAACAGCCACAACGTCTATATCATGGCCGACAAGCAGAAGAACGGCA
TCAAGGTGAACTTCAAGATCCGCCACAACATCGAGGGCGGCAGCGTGCAGCTCGCCGACCAC
TACCAGCAGAACACCCCCATCGGCGACGGCCCCGTGCTGCTGCCCGACAACCACTACCTGAGC
TACCAGTCCGCCCTGAGCAAAGACCCCAACGAGAAGCGCGATCACATGGTCCTGCTGGAGTTCG
TGACCGCCGCCGGGATCACTCTCGGCATGGACGAGCTGTACAAG

*eyfp* in pCQ11-FtsZ-eYFP
ATGGTTTCAAAAGGTGAAGAATTATTCACAGGTGTTGTTCCAATTTTGGTTGAATTAGATGGTGA
TGTTAATGGTCATAAAATTCTCAGTTTCAGGTGAAGGTGAAGGTGATGCAACATATGGTAAA
TTAACATTAAAATTTATTTGTACAACAGGTAAATTACCAGTTCCTTGGCCAACATTAGTTACAACA
TTCGGTTATGGTTTACAATGTTTTGCACGTTATCCAGATCATATGAAACAACATGATTTTTTTCAAA
TCAGCAATGCCTGAAGGTTATGTTCAAGAACGTACAATTTTCTTTAAAGATGATGGTAATTA-
CAAAACACGTGCTGAAGTGAAATTTGAAGGTGATACATTAGTTAATCGTATTGAATTAAAAGGTA
TTGATTTTAAAGAAGATGGAAATATTTTAGGTCATAAATTAGAATATAATTATAATTCACATAATG
TTTATATTATGGCAGATAAACAAAAAAATGGTATTAAAGTTAATTTCAAAATTCGTCATAATA
TTGAAGGTGGTTCAGTTCAATTAGCAGATCATTATCAACAAAATACACCTATTGGTGATGGTCCAG
TTTTATTACCAGATAATCATTATTTATCATATCAATCAGCATTATCAAAAGATCCAAATGAAAAACG
TGATCATATGGTTTTATTAGAATTTGTTACAGCAGCAGGTATTACATTAGGTATGGATGAATTATA
TAAATAA

*gfp* in pKASBAR-EzrA-GFP
ATGGCTAGCAAAGGAGAAGAACTTTTCACTGGAGTTGTCCCAATTCTTGTTGAATTAGATGGTGA
TGTTAATGGGCACAAATTTTCTGTCAGTGGAGAGGGTGAAGGTGATGCTACATACGGAAAGC
TTACCCTTAAATTTATTTGCACTACTGGAAAACTACCTGTTCCATGGCCAACACTTGTCACTAC
TTTGACCTATGGTGTTCAATGCTTTTCCCGTTATCCGGATCATATGAAACGGCATGACTTTTTCAA-
GAGTGCCATGCCCGAAGGTTATGTACAGGAACGCACTATATCTTTCAAAGATGACGGGAAC
TACAAGACGCGTGCTGAAGTCAAGTTTGAAGGTGATACCCTTGTTAATCGTATCGAGTTAAAAGG
TATTGATTTTAAAGAAGATGGAAACATTCTCGGACACAAACTCGAGTACAACTATAACTCACACAA
TGTATACATCACGGCAGACAAACAAAAGAATGGAATCAAAGCTAACTTCAAAATTCGCCACAACA
TTGAAGATGGATCCGTTCAACTAGCAGACCATTATCAACAAAATACTCCAATTGGCGATGGCCC
TGTCCTTTTACCAGACAACCATTACCTGTCGACACAATCTGCCCTTTCGAAAGATCCCAAC-
GAAAAGCGTGACCACATGGTCCTTCTTGAGTTTGTAACTGCTGCTGGGATTACACATGGCATGGA
TGAGCTCTACAAATAA

*snap* in pSNAP-tag (T7)−2 and pKASBAR-EzrA-SNAP
ATGGACAAAGACTGCGAAATGAAGCGCACCACCCTGGATAGCCCTCTGGGCAAGCTGGAACTG
TCTGGGTGCGAACAGGGCCTGCACCGTATCATCTTCCTGGGCAAAGGAACATC
TGCCGCCGACGCCGTGGAAGTGCCTGCCCCAGCCGCCGTGCTGGGCGGACCAGAGCCACTGA
TGCAGGCCACCGCCTGGCTCAACGCCTACTTTCACCAGCCTGAGGCCATCGAGGAGTTCCCTG
TGCCAGCCCTGCACCACCCAGTGTTCCAGCAGGAGAGCTTTACCCGCCAGGTGCTGTGGAAAC
TGCTGAAAGTGGTGAAGTTCGGAGAGGTCATCAGCTACAGCCACCTGGCCGCCC
TGGCCGGCAATCCCGCCGCCACCGCCGCCGTGAAAACCGCCCTGAGCGGAAATCCCGTGCCCA
TTCTGATCCCCTGCCACCGGGTGGTGCAGGGCGACCTGGACGTGGGGGGCTACGAGGGCGGGC
TCGCCGTGAAAGAGTGGCTGCTGGCCCACGAGGGCCACAGACTGGGCAAGCCTGGGCTGGGT

*snap* in pSS26b, pSS26bFtsZ-C and pCQ11-FtsZ-SNAP
ATGGACAAAGATTGCGAAATGAAACGTACCACCCTGGATAGCCCGCTGGGCAAACTGGAAC
TGAGCGGCTGCGAACAGGGCCTGCATGAAATTAAACTGCTGGGTAAAGGCAC-
CAGCGCGGCCGATGCGGTTGAAGTTCCGGCCCCGGCCGCCGTGCTGGGTGGTCCGGAACCGC
TGATGCAGGCGACCGCGTGGCTGAACGCGTATTTTCATCAGCCGGAAGCGATTGAAGAA
TTTCCGGTTCCGGCGCTGCATCATCCGGTGTTTCAGCAGGAGAGCTTTACCCGTCAGGTGCTG
TGGAAACTGCTGAAAGTGGTTAAATTTGGCGAAGTGATTAGCTATCAGCAGCTGGCGGCCC
TGGCGGGTAATCCGGCGGCCACCGCCGCCGTTAAAACCGCGCTGAGCGGTAACCCGGTGCCGA
TTCTGATTCCGTGCCATCGTGTGGTTAGCTCTAGCGGTGCGGTTGGCGGTTATGAAGGTGGTC
TGGCGGTGAAAGAGTGGCTGCTGGCCCATGAAGGTCATCGTCTGGGTAAACCGGGTCTGGGA
TGA

Linker A
TCAGGTTCAGGTTCAGGTGGGCGCGCCTCAGGTTCAGGTTCAGGT

Linker B
GAATTCCCCATGGGTTCAGGTGGTGGTGGTTCA

## Labelling *S. aureus* with DAAs

DAAs were prepared by published methods (*Adams and Errington, 2009*; *Steele et al., 2011*; *Strauss et al., 2012*) or by modified procedures described in Appendix. ADA was obtained from Iris Biotech. These were incubated with mid-exponential phase ($OD_{600}$ ~0.3 to 0.4) *S. aureus* at 500 µM (1 mM for ADA-DA) and incubated on a rotary shaker at 37°C for the required labelling time. Samples were imaged using widefield microscopy, 3D-SIM or localisation microscopy as required. For 15 s labelling DAAs were used at 10 mM, 1 ml samples were mixed briefly by vortexing and fixed by addition of 500 µl 8% (w/v) ice-cold paraformaldehyde immediately after vortexing.

## Click chemistry

DAAs containing an azide functional group (ADA and ADA-DA) required chemical attachment of a fluorophore via the Click reaction (copper (I)-catalysed alkyne-azide cycloaddition). This was carried out using the Click-iT Cell Reaction Buffer Kit (ThermoFisher) as per the manufacturer's protocol. Alkyne dyes were added at 5 µg ml$^{-1}$.

## Labelling *S. aureus* with fluorescent vancomycin

Fixed cells were resuspended in PBS containing fluorescent vancomycin at 2 µM (prepared using succinimidyl ester of Amersham Cy3B (GE Healthcare) as previously described (*Daniel and Errington, 2003*). Samples were protected from light and incubated at room temperature for 30 min then washed by centrifugation and resuspension in water. For dual labelled samples, cells were labelled with required DAA as described above and fixed with 4% (w/v) paraformaldehyde prior to labelling with fluorescent vancomycin.

### Labelling *S. aureus* with NHS ester

*S. aureus* grown to mid-exponential phase (OD$_{600}$ ~0.5) were resuspended in PBS containing Alexa Fluor 647 NHS ester (Invitrogen) at 8 µg ml$^{-1}$ and incubated at room temperature for 5 min. Cells were then washed by centrifugation and resuspension in PBS.

### Labelling *S. aureus* with SNAP-Cell TMR-Star

*S. aureus* grown to mid-exponential phase (OD$_{600}$ ~0.5) were incubated with SNAP-Cell TMR-Star (New England Biolabs) at 500 nM for widefield microscopy or 3 µM for SIM at 37°C for 15 min. Cells were washed by centrifugation and resuspension in PBS.

### Fixing

With the exception of Slimfield microscopy which involved no fixation and 15 s DAA labelling which used 8% (w/v) ice-cold paraformaldehyde, all samples were fixed with 4% (w/v) paraformaldehyde prior to imaging.

### Widefield epifluorescence microscopy

Fixed cells were mounted on poly-L-Lysine coated slides and imaged on a Nikon Ti Inverted microscope fitted with a Lumencor Spectra X light engine. Images were taken using a 100x PlanApo (1.4 NA) oil objective using 1.518 RI oil and detected by an Andor Zyla sCMOS camera.

### OMX microscopy

Coverslips (High-precision, No.1.5H, 22 × 22 mm, 170 ± 5 µm, Marienfeld) were sonicated for 15 min in 1 M KOH, washed with water and incubated in poly-L-Lysine solution for 30 min. Coverslips were then further washed and dried with nitrogen. Fixed cells were then dried onto the coverslips with nitrogen and mounted on slides with ~5 µl Slow Fade Diamond (Invitrogen).

Structured Illumination Microscopy was carried out using a v4 DeltaVision OMX 3D-SIM system fitted with a Blaze module (Applied Precision, GE Healthcare, Issaquah, USA). Samples were illuminated using laser illumination. For each z slice, samples were imaged in five phase shifts and three angles, z-steps were 0.125 nm. Reconstructions were performed with the Softworx software (GE Healthcare) using OTFs optimised for the specific wavelength and oil used. The same software was used for deconvolution.

### Sample preparation for localisation microscopy

For all samples coverslips were prepared as for 3D-SIM Microscopy. All samples except for eYFP/meYFP and were mounted on slides with 5 µl GLOX buffer (0.5 mg ml$^{-1}$ glucose oxidase, 40 µg ml$^{-1}$ catalase, 10% (w/v) glucose in 50 mM Tris-HCl containing 10 mM NaCl (pH 8.0) containing either 10 or 100 mM mercaptoethylamine (MEA).

For eYFP/meYFP imaging (single colour) samples were mounted in 5 µl PLOX buffer (5 U ml$^{-1}$ pyranose oxidase, 40 µg ml$^{-1}$ catalase, 10% (w/v) glucose in 50 mM Tris-HCl, 10 mM NaCl (pH 8.0) prepared in heavy water (*Ong et al., 2015*).

For eYFP/Alexa Fluor 647 imaging (two-colour) samples were mounted in 5 µl PLOX containing 50 mM MEA. Where required, coverslips were sparsely coated with TetraSpeck beads (0.1 µm, Molecular Probes) prior to the application of cells.

### Bespoke localisation microscope

Localisation microscopy was carried out as previously described (*Huang et al., 2010*; *Turner et al., 2013*), but using OBIS 405 (50 mW) and OBIS 647 (120 mW) lasers, a 662 nm dichroic and a 676 (*Daniel and Errington, 2003*) nm emission filter. Calibration data for 3D reconstructions was obtained by recording images of fiducial particles while stepping the objective piezo.

### Nikon N-STORM localisation microscope

Localisation microscopy was carried out using a Nikon Ti-NS N-STORM version 1 with 3D capability in continuous mode. Objective used was a SR Apo TIRF 100x NA 1.49 and images detected using EMCCD camera (Andor DU-897) using the 17 MHz 16 bit mode with an EM Multiplier Gain of 300 and a conversion gain of 3. Calibration data for 3D reconstructions was obtained by recording

images of fiducial particles using the calibration mode. Custom-made filter cubes were used for eYFP/meYFP (no excitation filter, 488 nm dichroic, 525/50 nm emission) and two-colour imaging (red/far red; no excitation filter, multi-band dichroic with transmission at 410–480 nm, 500–550 nm, 570–630 nm and above 650 nm, multi-band emission with transmission at 570–620 nm and above 660 nm) imaging and the N-STORM cube for single colour Alexa Fluor 647 imaging. Imaging was done under oblique illumination but not full TIRF. Two colour eYFP and Alexa Fluor 647 imaging was performed using separate filter cubes whereas two colour imaging using Cy3B and Alexa Fluor 647 was performed using a single cube, as specified.

## Image reconstruction

Images were reconstructed as previously described (*Huang et al., 2008*) using either custom Matlab scripts (available at this reference [*Turner and Foster, 2018*]), the ThunderSTORM ImageJ/Fiji plugin (*Ovesný et al., 2014*) or Nikon elements software. All of these methods identify the locations of molecules by fitting Gaussian functions to regions of source data, and all yielded similar results.

Two colour data (where using a single multi-band filter cube) was reconstructed and aligned (registered) using Nikon elements. In summary, alignment is achieved by obtaining calibration images of the same fluorescent beads in both channels. The software then determines the way in which localisations in one channel must be offset to align with the other, based on the offsets in the apparent positions of the beads.

For two colour eYFP/Alexa Fluor 647 NHS ester imaging, using two filter cubes, the average position of a TetraSpeck fiducial was determined in both channels and a translational offset calculated for each image. This was applied to the Alexa Fluor 647 channel to approximately align the data. Whilst more sophisticated co-alignment methods exist, this was sufficient for us to draw the qualitative conclusions necessary for this part of our study.

## Image rendering

Images were rendered as 2D histograms using the ThunderSTORM ImageJ/Fiji plugin (*Ovesný et al., 2014*). Unless otherwise stated images were projected onto a single plane and the reconstructed pixel size was 10 nm. Semi-quantitative Matlab contour plots were used in some instances for ease of visualisation of key features in 3D reconstructions both on screen and in print. eYFP and eYFP/Alexa Fluor 647 NHS ester dual colour images were reconstructed with a pixel size of 5 nm with a Gaussian blur of 20 nm applied to make them easier to see.

## Analysis of localisation microscopy data

Ring-like groups of localisations were manually selected from fields. The centre and radius of a circle that best fit the points was then determined allowing the localisations to be represented using polar co-ordinates. Histograms of localisations with respect to angle (2° bin size) and distance from the centre of the circle (10 nm bin size) were then generated. The angular histograms were auto-correlated to test for the presence of similarly sized large groups of molecules which would create peaks or a very slow decay from 0° in the resulting graph. The distance histograms were plotted and compared with those resulting from simulations.

An additional, similar, analysis was carried modelling the septal shape as an ellipse (*Figure 1—figure supplement 3*).

## Simulation of localisation microscopy data

We used the simplest possible methods to simulate data to compare with that acquired on the microscope. Localisations were randomly distributed by angle on circles of a fixed radius. Localisation error comes from several physical sources, but was simulated by adding offsets in x and y taken independently and at random from a normal distribution of a defined standard deviation.

## Slimfield microscopy: Microscope setup

A bespoke single-molecule microscope was used, constructed around the body of a Zeiss inverted microscope with a 100 × 1.49 numerical aperture oil immersion total internal reflection fluorescence (TIRF) objective lens (Olympus) and an xyz nano positioning stage (Nanodrive, Mad City Labs). A 20 mW Obis 514 nm laser expanded to 10 μm full width at half maximum was used to excite meYFP

fluorescence combined with a dual pass CFP/YFP dichroic mirror with 20 nm transmission windows centred on 440 nm and 514 nm. A high-speed camera (Andor iXon DV860-BI) was used to image at 5 ms/frame with magnification at 50 nm/pixel. Data were acquired using custom LabView software.

## Slimfield microscopy: Sample preparation and imaging

*S. aureus* SH4604 (*ezrA-meyfp ΔezrA*) cells were imaged by immobilising them on an agarose pad suffused with media. These were constructed by placing a gene frame (Life Technologies) on a BK7 glass microscope slide (Fisher) and filling with ~500 µl 1% (w/v) agarose containing media. Once set, 5 µl of cell culture was spotted over the agarose and covered with a plasma cleaned coverslip.

## Slimfield microscopy: Image analysis

Cell bodies and apparent EzrA rings were segmented as outlined previously (*Wollman et al., 2016*). In brief, the cell body was found by segmenting both a five frame average EzrA-meYFP fluorescence and brightfield image using a threshold set by the background peak in the pixel intensity distribution. The brightfield segmentation was used as seeds for watershedding the segmented fluorescence image to identify individual cells. Further thresholding within cell pixels yields a mask for the EzrA ring.

Diffraction-limited fluorescent foci were tracked using custom Matlab software as described previously (*Wollman et al., 2015*). In brief, in each frame, candidate foci are identified by thresholding top-hat transformed images using Otsu's method. The spot centre is determined to sub-pixel precision using iterative Gaussian masking (*Leake et al., 2006*) and accepted if its signal-to-noise ratio, as defined by the foci intensity, the background-corrected integrated pixel intensity within a five pixel radius circular region of interest centred of the foci intensity centroid, divided by the standard deviation of the background pixels, is greater than 0.4. Foci are linked into the same track between image frames if they are within a distance of 1 optical resolution width (approximately five pixels), generating single particle tracks to a typical localization precision of ~40 nm (*Llorente-Garcia et al., 2014*).

The mean squared displacement of each track over its first four time interval points was used to calculate its microdiffusion coefficient, D, using a linear fit (*Kusumi et al., 1993*). These were binned into 0.01 µm$^2$ s$^{-1}$ bins and fitted with 1–3 gamma functions (*Stracy et al., 2015*), with three gammas generating the lowest reduced chi$^2$.

Copy number values were calculated using a deconvolution method called CoPro (*Wollman and Leake, 2015*) which utilised the symmetrical geometry of *S. aureus* cells and the in vivo characteristic intensity of single meYFP molecules (*Leake, 2014*). Detection of single meYFP was confirmed by observation of single, distinct photobleach steps. This characteristic brightness value corresponding to a single meYFP molecule was determined as the peak of the intensity distribution of fluorescent foci found after 200 ms of photobleaching, and was equivalent to 2000 ± 500 counts on our EMCCD camera detector.

## Transmission electron microscopy

Samples were prepared for electron microscopy as previously described (*Bottomley et al., 2014*).

## Cell volume calculation

Cell volume calculations were carried out as previously described (*Zhou et al., 2015*), specifically, the long and short axis of cells were measured using Fiji. The volume was then calculated based on a prolate spheroid shape with volume $V = \frac{4}{3}\pi ab^2$, where a and b are the dimensions of the long and short axis respectively.

## Gel-based analysis of SNAP tagged proteins

SNAP-Cell TMR-Star (New England Biolabs) was added to a 1 ml aliquot of mid-exponential phase (OD$_{600}$ ~1) grown culture at a concentration of 500 nM and incubated at 37°C for 1 hr. Cells were washed three times by resuspension and centrifugation in PBS, resuspended in PBS supplemented with 200 µg ml$^{-1}$ lysostaphin and 20 U ml$^{-1}$ DNase I and lysed at 37°C for 30 min. Cell extracts were resolved in SDS-PAGE, the gel was rinsed with dH$_2$O and scanned using ChemiDoc MP System (Bio-Rad).

## Incorporation of $^{14}$C-GlcNAc into cell wall peptidoglycan

*S. aureus* strains were grown overnight in CDM and used to inoculate fresh CDM to an $OD_{600}$ of 0.05 and grown to $OD_{600} \sim 0.2$. At this point 5 µM $^{14}$C-GlcNAc was added to cultures. At 30 min intervals samples were collected and prepared for analysis of $^{14}$C-GlcNAc incorporation via Liquid Scintillation as previously described (*Maki et al., 2001*).

## Fluorescence intensity measurements

Fluorescence intensity was measured using Image J/Fiji and calculated as counts/pixel. To determine the % off-septal fluorescence the fluorescence intensity for both the septum and the whole cell was measured and the percentage of non-septal fluorescence calculated.

## Peptidoglycan purification and mass-spectrometry analysis

*S. aureus* peptidoglycan was purified as previously described (*Turner et al., 2010*). Specifically, 1L cultures of *S. aureus* SH1000 and *S. aureus* SH1000 containing 1 mM ADA were grown for 4 hr before peptidoglycan was extracted and purified. Peptidoglycan was solubilized by digestion with 50 µg Cellosyl per mg peptidoglycan (dry weight) overnight at 37°C. Samples were boiled to inactivate the Cellosyl and reduced using sodium borohydride (*Bern et al., 2017*). Reduced muropeptides were separated on an Agilent Technologies Accurate Mass Q-TOF LC/MS using a Hypersil Gold aQ column (200 × 42.1 µm, 1.9 µm particle size) with a gradient of 0–30% (v/v) water/ACN both containing 0.1% (v/v) formic acid over 60 mins.

## Acknowledgements

This work was funded by the Medical Research Council (MR/N002679/1, MR/K015753/1, G1100127, MR/K01580X/1) and the Biotechnology and Biological Science Research Council UK (BB/L006162/1, BB/N006453/1). We are grateful to Gareth McVicker, Simon Thorpe, Chris Hill, Irene Johnson and Joe Kirk for their assistance.

## Additional information

### Funding

| Funder | Grant reference number | Author |
| --- | --- | --- |
| Medical Research Council | MR/N002679/1 | Simon J Foster |
| Biotechnology and Biological Sciences Research Council | BB/L006162/1 | Simon J Foster |
| Medical Research Council | MR/K015753/1 | Simon J Foster |
| Medical Research Council | G1100127 | Simon J Foster |
| Medical Research Council | MR/K01580X/1 | Mark C Leake |
| Biotechnology and Biological Sciences Research Council | BB/N006453/1 | Mark C Leake |

The funders had no role in study design, data collection and interpretation, or the decision to submit the work for publication.

### Author contributions

Victoria A Lund, Katarzyna Wacnik, Robert D Turner, Conceptualization, Data curation, Formal analysis, Investigation, Methodology, Writing—original draft, Writing—review and editing; Bryony E Cotterell, Conceptualization, Data curation, Formal analysis, Validation, Investigation, Methodology, Writing—original draft, Writing—review and editing; Christa G Walther, Resources, Formal analysis, Investigation, Methodology, Writing—review and editing; Samuel J Fenn, Formal analysis, Investigation, Methodology, Writing—review and editing; Fabian Grein, Data curation, Methodology; Adam JM Wollman, Resources, Data curation, Methodology; Mark C Leake, Simon Jones, Formal analysis, Methodology, Writing—review and editing; Nicolas Olivier, Conceptualization, Supervision, Funding

acquisition, Project administration, Writing—review and editing; Ashley Cadby, Methodology, Writing—review and editing; Stéphane Mesnage, Supervision, Methodology, Writing—review and editing; Simon J Foster, Formal analysis, Supervision, Funding acquisition, Project administration, Writing—review and editing

### Author ORCIDs
Victoria A Lund http://orcid.org/0000-0002-1637-2023
Katarzyna Wacnik http://orcid.org/0000-0002-9921-6746
Christa G Walther https://orcid.org/0000-0002-8962-3102
Adam JM Wollman http://orcid.org/0000-0002-5501-8131
Mark C Leake http://orcid.org/0000-0002-1715-1249
Simon Jones http://orcid.org/0000-0001-8043-7998
Simon J Foster https://orcid.org/0000-0001-7432-7805

### Decision letter and Author response
Decision letter https://doi.org/10.7554/eLife.32057.040
Author response https://doi.org/10.7554/eLife.32057.041

## Additional files

### Supplementary files
• Transparent reporting form
DOI: https://doi.org/10.7554/eLife.32057.015

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

# Appendix 1

DOI: https://doi.org/10.7554/eLife.32057.016

## General synthetic methods

H-D-aza-OH.HCl, T3P in DMF, H-D-ala-O$^t$Bu, *N*-succinimidyl-7-hydroxycoumarin-3-carboxylate, Boc-D-Dap-OH, 6-[tetramethylrhodamine-5(6)-carboxamido]hexanoic acid succinimidyl ester are commercially available and were used as supplied.

## Synthesis of 3-azido-*N*-[(1,1-dimethylethoxy)carbonyl]-D-alanine (Boc-D-aza-ala-OH) (*Appendix 1—figure 1*)

H-D-aza-OH.HCl (527 mg, 4.1 mmol) was added to a stirred solution of sodium carbonate (1.17 g, 11.1 mmol) in water (5 mL). The solution was cooled to 0°C, followed by drop-wise addition of a solution of di-*t*-butyldicarbonate (1.25 g, 5.3 mmol) in acetonitrile (5 mL). The mixture was allowed to warm to room temperature overnight, diluted with water (50 mL), and washed with diethyl ether (20 mL). The aqueous layer was acidified to pH 2 using 1M HCl and extracted with ethyl acetate (3 × 50 mL). The combined organic layers were washed with brine (20 mL), dried over magnesium sulfate, filtered and evaporated to give a clear, colourless oil (850 mg, 90%) that was used directly without further purification; $[\alpha]_D^{25}$ −25.0 (*c* 1.0, MeOH); $\nu_{max}$/cm$^{-1}$ (film) 3331, 2981, 2935, 2551, 2108, 1714, 1513; $^1$H NMR (400 MHz, MeOD) $\delta_H$ 4.33 (1H, t *J* 5.1 Hz, C*H*), 3.70–3,63 (2H, m, C*H$_2$*), 1.48 [9H, s, (C*H$_3$*)$_3$] (*Appendix 1—figure 2*); $^{13}$C NMR (100 MHz, MeOD) $\delta_C$ 171.5 (CO), 156.3 (CO), 79.5 (*C*), 53.5 (CH), 51.8 (CH$_2$), 27.2 (CH$_3$) (*Appendix 1—figure 3*); *m/z* (TOF MS ES$^+$) 365 (30%), 297 (*Komis et al., 2015*), 229.0949 (100, M$^+$-H. C$_8$H$_{13}$N$_4$O$_4$ requires 229.0942).

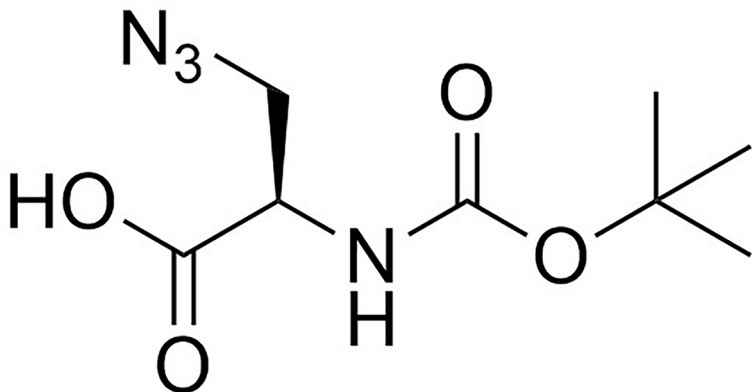

**Appendix 1—figure 1.** Structure of 3-Azido-*N*-[(1,1-dimethylethoxy)carbonyl]-D-aza-ala-OH).

DOI: https://doi.org/10.7554/eLife.32057.017

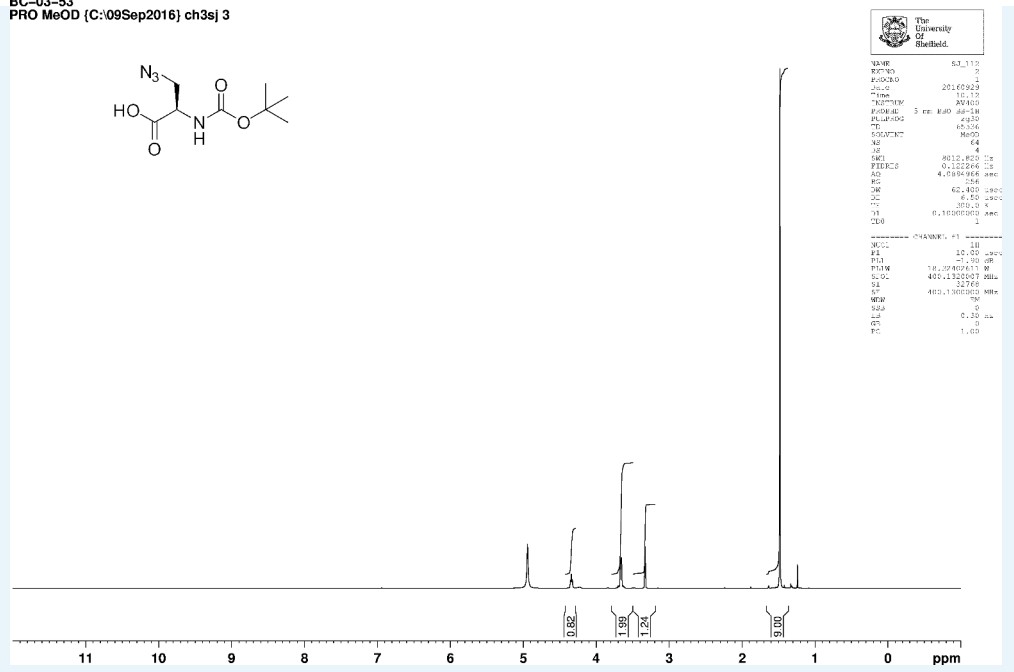

**Appendix 1—figure 2.** 1D ¹H NMR Spectrum of 3-Azido-*N*-[(1,1-dimethylethoxy)carbonyl]-D-alanine (Boc-D-aza-ala-OH.

DOI: https://doi.org/10.7554/eLife.32057.018

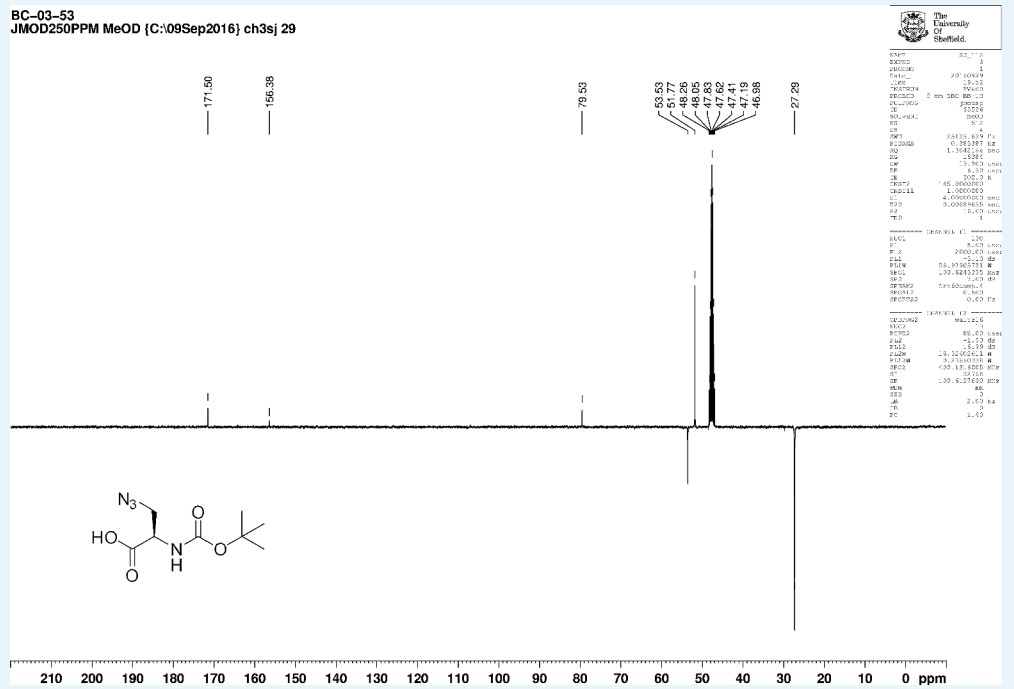

**Appendix 1—figure 3.** 1D ¹³C NMR Spectrum of 3-Azido-*N*-[(1,1-dimethylethoxy)carbonyl]-D-alanine (Boc-D-aza-ala-OH).

DOI: https://doi.org/10.7554/eLife.32057.019

This compound has previously been prepared in the literature by different methods. NMR data have been reported in DMSO – no specific rotation was recorded (*Aggen et al., 1999*).

# Synthesis of 3-azido-*N*-[(1,1-dimethylethoxy)carbonyl] -D-alanyl-D-alanine 1,1-dimethylethyl ester (Boc-D-aza-ala-D-ala-OᵗBu) (*Appendix 1—figure 4*)

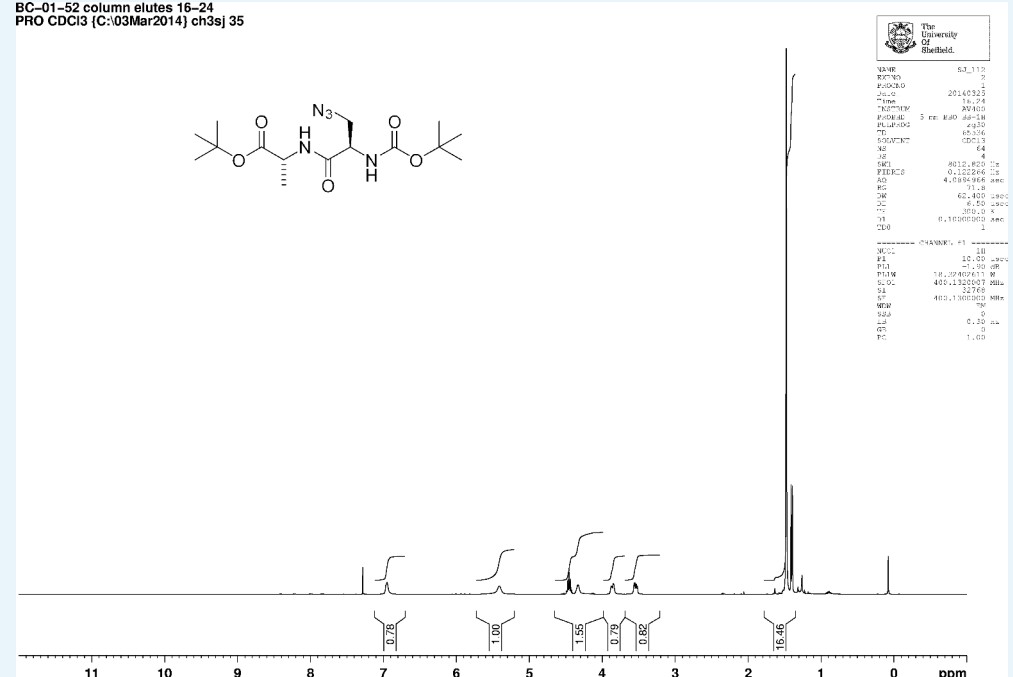

**Appendix 1—figure 4.** Structure of 3-Azido-*N*-[(1,1-dimethylethoxy)carbonyl]-D-alanyl-D-alanine 1,1-dimethylethyl ester (Boc-D-aza-ala-D-ala-OtBu).
DOI: https://doi.org/10.7554/eLife.32057.020

Boc-D-aza-ala-OH (850 mg, 3.7 mmol), *N*-methylmorpholine (4.5 mL, 41 mmol) and H-D-ala-OᵗBu (710 mg, 3.7 mmol) were added to dry DCM (100 mL) under argon. A 50 wt% solution of T3P in DMF (2.4 g, 5.0 mL, 7.5 mmol) was added slowly and the mixture was stirred at room temperature for 18 hr. The solvent was removed under vacuum, the remaining solid purified by column chromatography on silica, eluting with a 1% to 5% gradient of methanol in dichloromethane, to give a colourless oil (1.07 g, 73%); $[\alpha]_D^{22}$+5.45 (*c* 1.0, CHCl₃); $v_{max}/cm^{-1}$ (ATR) 3020, 2986, 2930, 2104, 1721, 1661; ¹H NMR (400 MHz, CDCl₃) $\delta_H$6.95 (1H, br d, *J* 6.2 Hz, N*H*), 5.41 (1H, br s, N*H*), 4.45 (1H, quint, *J* 7.1 Hz, CH₃C*H*), 4.42 (1H, br s, C*H*) 3.86 (1H, br dd, *J* 12.2 Hz and 2.6 Hz, CH*H*), 3.54 (1H, dd, *J* 12.2 Hz and 5.4 Hz, CH*H*), 1.48 (9H, s, 3 × C*H₃*), 1.40 (3H, d, *J* 7.1 Hz, C*H₃*) (*Appendix 1—figure 5*); ¹³C NMR (100 MHz, CDCl₃) $\delta_C$ 173.9 (*C* = O), 165.9 (*C* = O), 52.3 (*C*H), 50.8 (*C*H₂), 48.5 (*C*H), 16.1 (*C*H₃) (*Appendix 1—figure 6*); *m/z* (TOF MS ES⁺) 358.2093 (100%, MH⁺. C₁₅H₂₈N₅O₅ requires 358.2090).

**Appendix 1—figure 5.** 1D ¹H NMR Spectrum of 3-Azido-*N*-[(1,1-dimethylethoxy)carbonyl]-D-alanyl-D-alanine 1,1-dimethylethyl ester (Boc-D-aza-ala-D-ala-OtBu).
DOI: https://doi.org/10.7554/eLife.32057.021

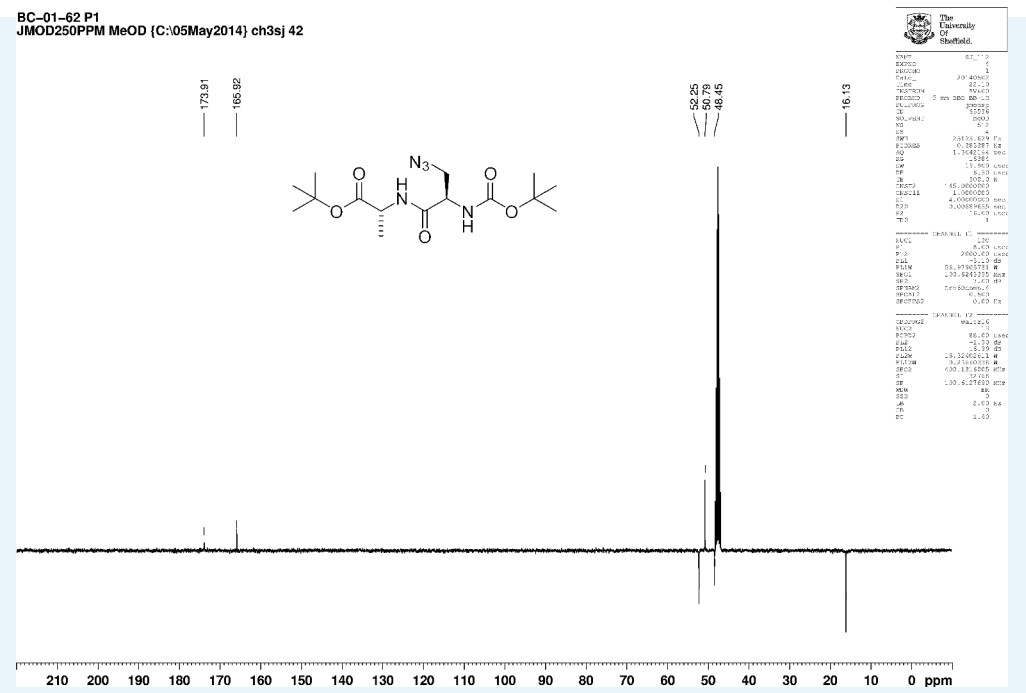

**Appendix 1—figure 6.** 1D $^{13}$C NMR Spectrum of 3-Azido-N-[(1,1-dimethylethoxy)carbonyl]-D-alanyl-D-alanine 1,1-dimethylethyl ester (Boc-D-aza-ala-D-ala-OtBu).

DOI: https://doi.org/10.7554/eLife.32057.022

# Synthesis of 3-azido-ᴅ-alanyl-ᴅ-alanine, 2,2,2-trifluoroacetate [ᴅ-aza-ala-ᴅ-ala (ADA-DA) TFA salt] (*Appendix 1—figure 7*)

**Appendix 1—figure 7.** Structure of 3-Azido-D-alanyl-D-alanine, 2,2,2-trifluoroacetate [D-aza-ala-D-ala (ADA-DA) TFA salt].

DOI: https://doi.org/10.7554/eLife.32057.023

Boc-ᴅ-aza-ala-ᴅ-ala-O$^t$Bu (112 mg, 0.28 mmol) stirred with TFA (1 mL) in methanol (5 mL) for 5 hr at room temperature. The solvent was removed *in vacuo* to give a colourless oil (83 mg, 100%); $[\alpha]_D^{23} -2.00$ (c 1.0, MeOH); $\nu_{max}$/cm$^{-1}$ (film) 3081, 2120, 1670, 1563; $^1$H NMR (400 MHz, MeOD) $\delta_H$ 4.46 (1H, q, *J* 7.3 Hz, CH$_3$C*H*), 4.07 (1H, dd, *J* 4.0 Hz and 7.5 Hz, CH$_2$C*H*) 3.96 (1H, dd, *J* 4.0 Hz and 13.5 Hz, C*H*H), 3.77 (1H, dd, *J* 7.5 Hz and 13.5 Hz, CH*H*), 1.46 (3H, d, *J* 7.3 Hz, CH$_3$) (*Appendix 1—figure 8*); $^{13}$C NMR (100 MHz, MeOD) $\delta_C$ 173.9 (CO), 165.9 (CO), 52.3 (CH), 50.8 (CH$_2$), 48.4 (CH), 16.1 (CH$_3$) (*Appendix 1—figure 9*); *m/z* (TOF MS ES$^+$) 224 (10%), 202.0940 (100, MH$^+$. C$_6$H$_{12}$N$_5$O$_3$ requires 202.0935), 145 (*Steele et al., 2011*).

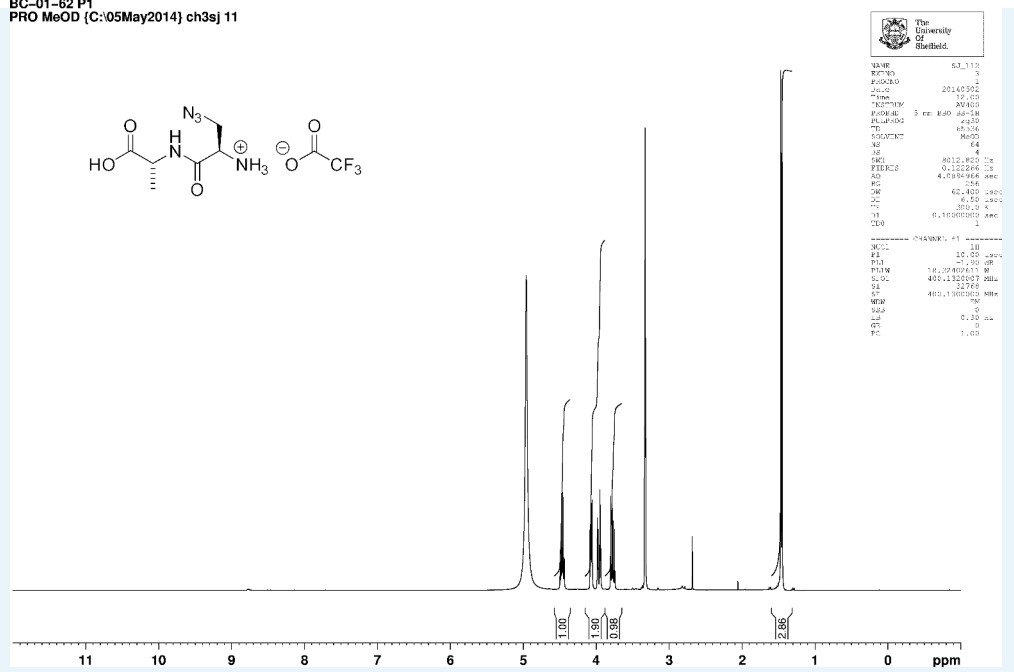

**Appendix 1—figure 8.** 1D [1]H NMR Spectrum of 3-Azido-D-alanyl-D-alanine, 2,2,2-trifluoroacetate [D-aza-ala-D-ala (ADA-DA) TFA salt].

DOI: https://doi.org/10.7554/eLife.32057.024

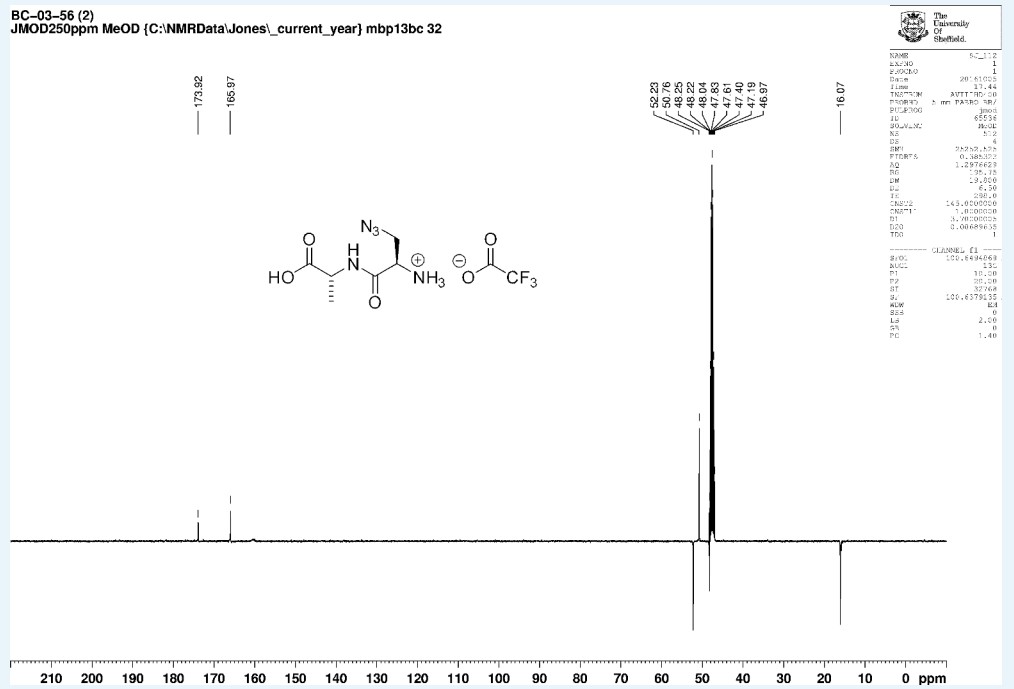

**Appendix 1—figure 9.** 1D [13]C NMR Spectrum of 3-Azido-D-alanyl-D-alanine, 2,2,2-trifluoroacetate [D-aza-ala-D-ala (ADA-DA) TFA salt].

DOI: https://doi.org/10.7554/eLife.32057.025

The compound has been previously reported as a white solid, although no specific rotation or IR data were reported. [1]H and [13]C literature date was recorded in DMSO, but is in broad agreement (*Liechti et al., 2014*).

# Synthesis of *N*-[(1,1-Dimethylethoxy)carbonyl]−3-{[(7-hydroxy-2-oxo-2*H*-1-benzopyran-3-yl) carbonyl] amino} D-alanine (N-Boc-HADA) (*Appendix 1—figure 10*)

**Appendix 1—figure 10.** Structure of *N*-[(1,1-Dimethylethoxy)carbonyl]−3-{[(7-hydroxy-2-oxo-2H-1-benzopyran-3-yl) carbonyl] amino} D-alanine (N-Boc-HADA).

DOI: https://doi.org/10.7554/eLife.32057.026

*N*-Succinimidyl-7-hydroxycoumarin-3-carboxylate (250 mg, 0.82 mmol) was dissolved in triethylamine (0.26 mL) and dry $CH_2Cl_2$ (15 mL) under an inert atmosphere at room temperature. Boc-D-Dap-OH (168 mg, 0.82 mmol) was added to the mixture and stirred for 18 hr at room temperature. The mixture was diluted with ethyl acetate (50 mL) and washed with 1M HCl (2 × 10 mL). The aqueous layer was extracted with ethyl acetate (3 × 10 ml) the combined organic layers dried over magnesium sulfate, then filtered. The solvent was removed *in vacuo* to yield white powder (317 mg, 99%) the powder could be further purified by recrystallization from methanol (186 mg, 58%); $[\alpha]_D^{25}$−54.0 (*c* 1.0, MeOH); $\nu_{max}$/cm$^{-1}$ (film) 3411, 1692, 1600, 1543; $^1$H NMR (400 MHz, MeOD) $\delta_H$8.79 (1H, s, ArC*H*), 7.69 (1H, d, *J* 8.6 Hz, ArC*H*), 6.91 (1H, dd, *J* 2.0 Hz and 8.6 Hz, ArC*H*), 6.78 (1H, d, *J* 2.0 Hz, ArC*H*), 4.42–4.39 (1H, m, C*H*), 3.99 (1H, dd, *J* 13.5 Hz and 4.6 Hz, C*H*H), 3.60 (1H, dd, *J* 13.5 and 8.2 Hz, CH*H*), 1.44 (9H, s, C*H₃*) (*Appendix 1—figure 11*); $^{13}$C NMR (100 MHz, MeOD) $\delta_C$ 172.8 (*CO*), 164.4 (*CO*), 163.6 (*CO*), 161.6 (*CO*), 157.0 (*C*), 156.5 (*C*), 148.5 (*CH*), 131.6 (*CH*), 114.3 (*CH*), 112.9 (*C*), 111.4 (*C*), 101.7 (*CH*), 79.3 (*C*), 53.7 (*CH*), 40.7 (*CH₂*), 27.3 (*CH₃*) (*Appendix 1—figure 12*); $m/z$ (TOF MS ES$^+$) 391.1166 (100%, M$^+$-H. $C_{18}H_{19}N_2O_8$ requires 391.1147).

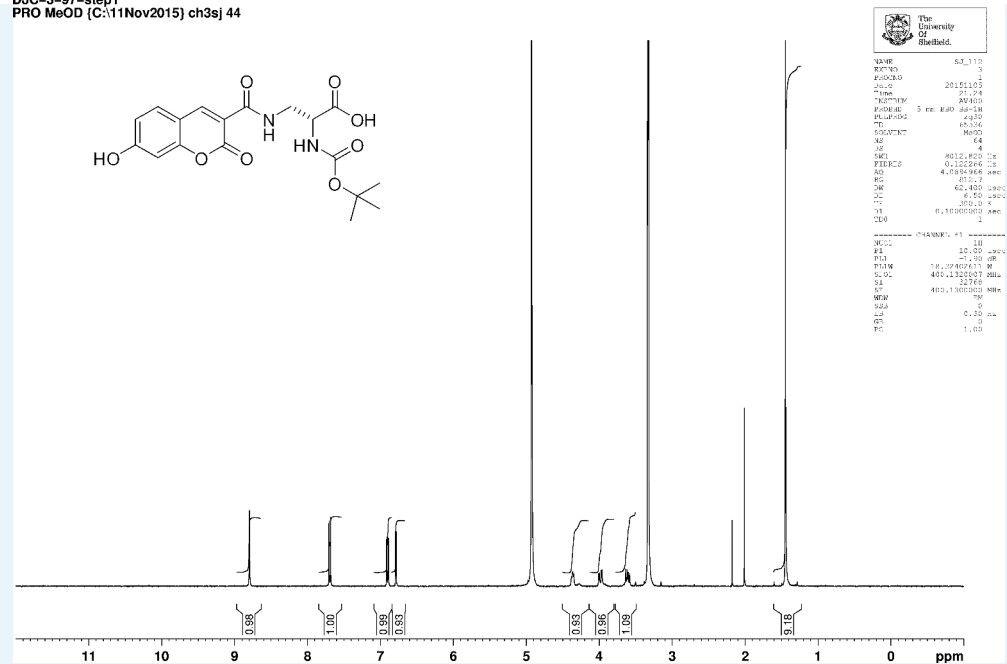

**Appendix 1—figure 11.** 1D $^1$H NMR Spectrum of *N*-[(1,1-Dimethylethoxy)carbonyl]−3-{[(7-hydroxy-2-oxo-2H-1-benzopyran-3-yl) carbonyl] amino} D-alanine (N-Boc-HADA).

DOI: https://doi.org/10.7554/eLife.32057.027

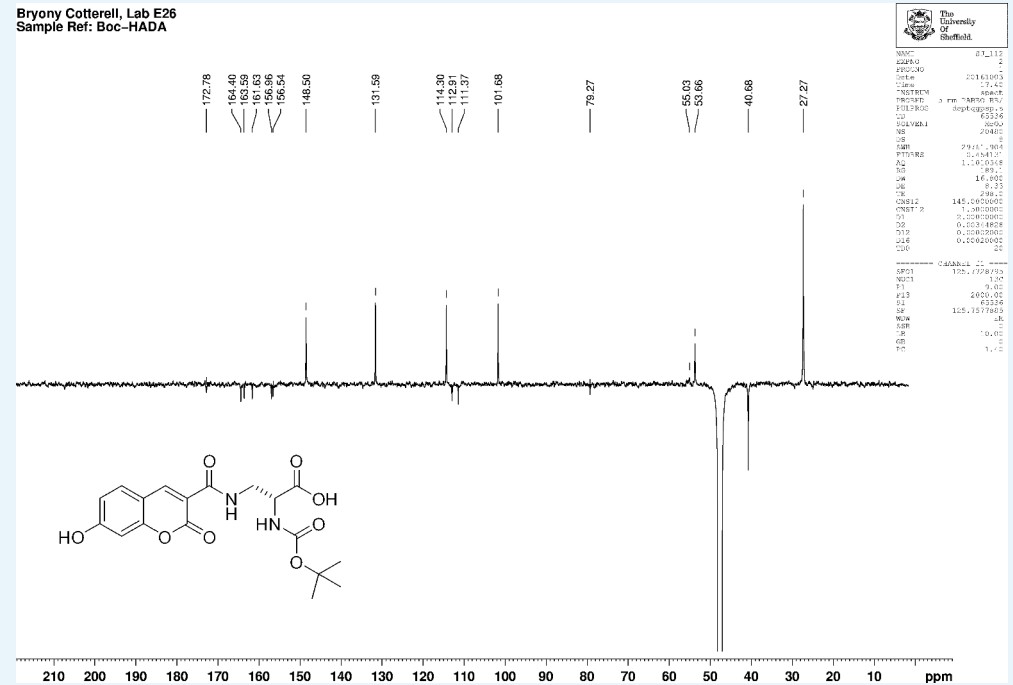

**Appendix 1—figure 12.** 1D $^{13}$C NMR Spectrum of *N*-[(1,1-Dimethylethoxy)carbonyl]−3-{[(7-hydroxy-2-oxo-2H-1-benzopyran-3-yl) carbonyl] amino} D-alanine (N-Boc-HADA).

DOI: https://doi.org/10.7554/eLife.32057.028

# Synthesis of 3-{[(7-Hydroxy-2-oxo-2*H*-1-benzopyran-3-yl) carbonyl] amino} D-alanine, 2,2,2-trifluoroacetate (HADA TFA salt) (*Appendix 1—figure 13*)

**Appendix 1—figure 13.** Structure of 3-{[(7-Hydroxy-2-oxo-2H-1-benzopyran-3-yl) carbonyl] amino} D-alanine, 2,2,2-trifluoroacetate (HADA TFA salt).

DOI: https://doi.org/10.7554/eLife.32057.029

Trifluoroacetic acid (2 mL) was added to N-Boc-HADA (96 mg, 0.24 mmol) in DCM (2 mL) and stirred at room temperature for 3 hr. The solvent was removed *in vacuo* to yield dark orange oil that was used without further purification (72 mg, 99%); $[\alpha]_D^{23} -1.5$ ($c$ 1.0, MeOH); $\nu_{max}/cm^{-1}$ (film) 3388, 3315, 3143, 2947, 2838, 1701, 1626, 1539, 1430; $^1$H NMR (400 MHz, MeOD), $\delta_H$ 8.81 (1H, s, ArC*H*), 7.69 (1H, d, *J* 8.6 Hz, ArC*H*), 6.91 (1H, dd, *J* 8.6 Hz and 2.0 Hz, ArC*H*), 6.78 (1H, d, *J* 2.0 Hz, ArC*H*), 4.29 (1H, dd, *J* 6.4 Hz and 3.8 Hz, C*H*), 3.99 (1H, dd, *J* 3.8 Hz and 14.5 Hz, C*H*H), 3.6 (1H, dd, *J* 6.4 Hz and 14.5 Hz, C*H*H) (*Appendix 1—figure 14*); $^{13}$C NMR (400 MHz, MeOD), $\delta_C$ 168.5 (*CO*), 164.7 (*CO*), 164.7 (*CO*), 161.7 (*C*), 157.1 (*C*), 148.9 (*CH*), 131.7 (*CH*), 114.4 (*CH*), 112.5 (*C*), 111.3 (*C*), 101.7 (*CH*), 53.2 (*CH*), 39.1 (*CH₂*) (*Appendix 1—figure 15*); *m/z* (TOF MS ES⁺) 305 (100%), 293.0777 (65, MH⁺. $C_{13}H_{13}N_2O_6$ requires 292.0774).

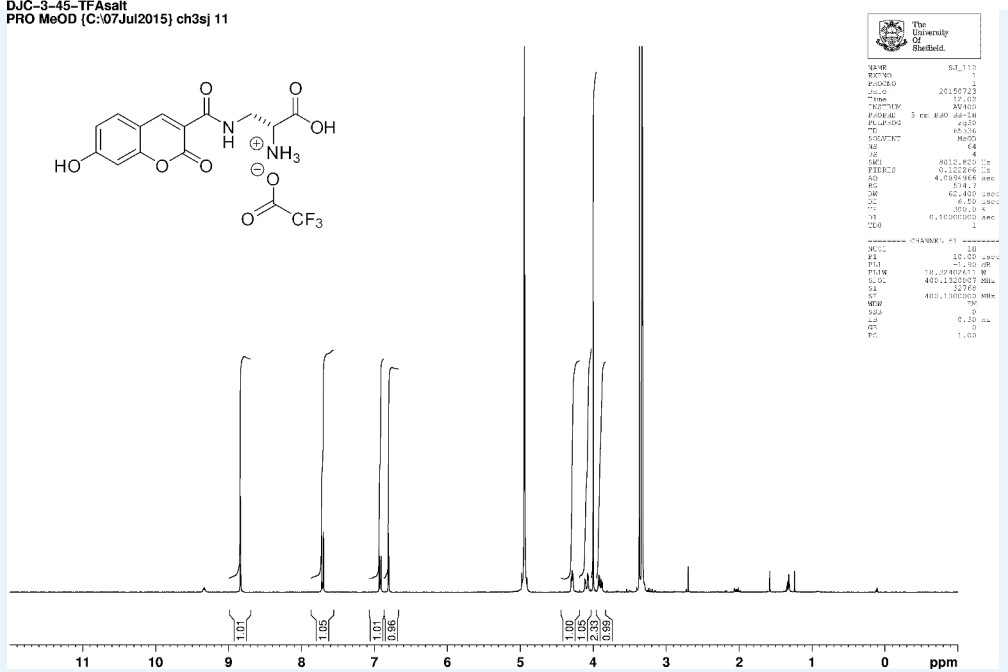

**Appendix 1—figure 14.** 1D [1]H NMR Spectrum of 3-{[(7-Hydroxy-2-oxo-2H-1-benzopyran-3-yl) carbonyl] amino} D-alanine, 2,2,2-trifluoroacetate (HADA TFA salt).

DOI: https://doi.org/10.7554/eLife.32057.030

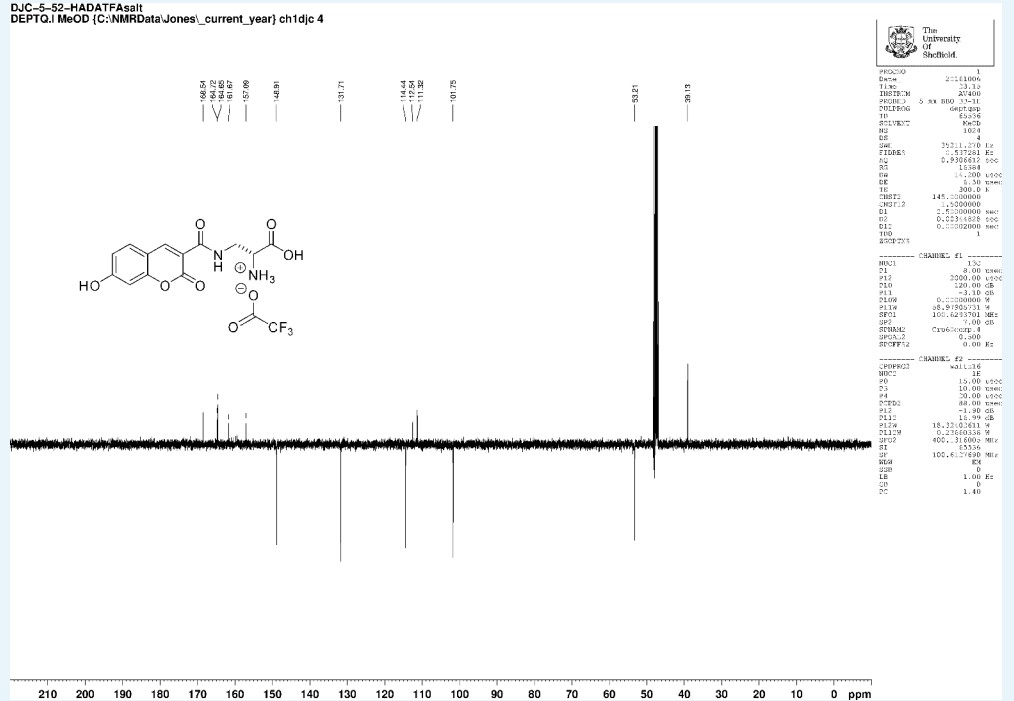

**Appendix 1—figure 15.** 1D [13]C NMR Spectrum of 3-{[(7-Hydroxy-2-oxo-2H-1-benzopyran-3-yl) carbonyl] amino} D-alanine, 2,2,2-trifluoroacetate (HADA TFA salt).

DOI: https://doi.org/10.7554/eLife.32057.031

The compound has been previously reported, but all analytical data have been recorded in DMSO, so is in broad agreement (*Kuru et al., 2012*).

**Appendix 1—table 1.** Strains used in this study.

| Strain | Relevant Genotype/markers | Source |
|---|---|---|
| SH1000 | Functional *rsbU*[+] derivative of 8325–4 | (*Horsburgh et al., 2002b*) |
| RN4220 | Restriction deficient transformation recipient | (*Kreiswirth et al., 1983*) |
| CYL316 | *S. aureus* RN4220 pCL112Δ19 (cm) | (*Lee et al., 1991*) |
| JGL227 | *S. aureus* SH1000 *ezrA-gfp+* (ery) | (*Steele et al., 2011*) |
| SH4386 | *S. aureus* SH1000 *ezrA-eyfp* (kan) | This study |
| SH4388 | *S. aureus* SH1000 *ezrA-eyfp ΔezrA* (kan, tet) | This study |
| SH4603 | *S. aureus* SH1000 *ezrA-meyfp* (kan) | This study |
| SH4604 | *S. aureus* SH1000 *ezrA-meyfp ΔezrA* (kan, tet) | This study |
| SH4639 | *S. aureus* SH1000 *ezrA-gfp* (kan) | This study |
| SH4640 | *S. aureus* SH1000 *ezrA-gfp ΔezrA* (kan, tet) | This study |
| SH4641 | *S. aureus* SH1000 *ezrA-snap* (kan) | This study |
| SH4642 | *S. aureus* SH1000 *ezrA-snap ΔezrA* (kan, tet) | This study |
| SH4652 | *S. aureus* SH1000 *ezrA-eyfp ΔezrA* pCQ11-FtsZ-SNAP (kan, tet, ery) | This study |
| SH4665 | *S. aureus* SH1000 pCQ11-FtsZ-eYFP (ery) | This study |
| NE679 | *S. aureus* JE2 with transposon insertion in *pbp4* (ery) | (*Fey et al., 2013*) |
| SH4425 | *S. aureus* SH1000 *pbp4* (ery) | This study |
| N315 | Methicillin-resistant *S. aureus* | (*Kuroda et al., 2001*) |
| SU492 | *B. subtilis* SU5 P$_{xyl}$-*ftsZ-yfp* (spec) | (*Monahan et al., 2014*) |

DOI: https://doi.org/10.7554/eLife.32057.032

**Appendix 1—table 2.** Plasmids used in this study

| Plasmid | Relevant genotype/markers | Source |
|---|---|---|
| pGM074 | pKASBAR-kan (*Bottomley et al., 2014*) carrying *ezra-psmorange* under the putative *ezrA* promoter (amp, kan) | G. McVicker |
| pSNAP-tag (T7)−2 | *E. coli* expression plasmid carrying the *snap* gene under the control of the T7 promoter (amp) | New England Biolabs |
| pOB | pGEM3Zf(+) cloning vector containing the erythromycin resistance cassette (amp, ery) | (*Horsburgh et al., 2002a*) |
| pAISH | TetR derivative of pMUTIN4 | (*Aish, 2003*) |
| pKASBAR-EzrA-eYFP | pKASBAR-kan containing *ezrA-eyfp* under the putative *ezrA* promoter (amp, kan) | This study |
| pKASBAR-EzrA-meYFP | pKASBAR-kan containing *ezrA-meyfp* under the putative *ezrA* promoter (amp, kan) | This study |
| pKASBAR-EzrA-GFP | pKASBAR-kan containing *ezrA-gfp* under the putative *ezrA* promoter (amp, kan) | This study |
| pKASBAR-EzrA-SNAP | pKASBAR-kan containing *ezrA-snap* under the putative *ezrA* promoter (amp, kan) | This study |
| pOB-ΔezrA | pOB containing the *ezrA* deletion cassette consisting of a 1.5 kb fragment of the upstream region of *S. aureus ezrA*, the tetracycline resistance cassette from pAISH and a 1.5 kb fragment of the downstream region of *S. aureus ezrA* (amp, ery, tet) | This study |
| pSS26b | pUC19 encoding *snap* (amp) | Covalys |
| pSS26bFtsZ-C | pSS26b containing *ftsZ-snap* (amp) | This study |
| pCQ11 | *E. coli-S. aureus* shuttle vector containing *lacI*, Pspac and *gfp* (amp, ery) | (*Hardt et al., 2017*) |

*Appendix 1—table 2 continued on next page*

*Appendix 1—table 2 continued*

| Plasmid | Relevant genotype/markers | Source |
|---|---|---|
| pCQ11-FtsZ-SNAP | pCQ11 derivative containing *ftsZ-snap* under Pspac (amp, ery) | This study |
| pCQ11-FtsZ-eYFP | pCQ11-FtsZ-SNAP with *eyfp* replacement of *snap* (amp, ery) | This study |

DOI: https://doi.org/10.7554/eLife.32057.033

**Appendix 1—table 3.** Oligonucleotides used in this study.

| Oligonucleotide name | Sequence (5' to 3') |
|---|---|
| eYFP-F | CGGCGCGCCTCAGGTTCAGGTTCAGGTATGGTGAGCAAGGGCGAG |
| eYFP-R | CGCGGCCGCTTACTTGTACAGCTCGTCCATGCCGAGAGTGATCCCGGC |
| GFP-F | CGGCGCGCCTCAGGTTCAGGTTCAGGTATGGCTAGCAAAGGAGAAGAACTTTTCACTGGAGTTGTCCC |
| GFP-R | CGCGGCCGCTTATTTGTAGAGCTCATCCATGCCATGTGTAATCCCAGCAGC |
| SNAP-F | GGGCGCGCCTCAGGTTCAGGTTCAGGTATGGACAAAGACTGCGAAATGAAGCGCAC |
| SNAP-R | CGAATTCTCATTAACCCAGCCCAGGCTTGCCCAGTCTG |
| meYFP-F | CTACCAGTCCAAGCTGAGCAAAGAC |
| meYFP-R | CTCAGGTAGTGGTTGTCG |
| pOB-ezrA-up-F | TTTACGTACACTATCTGCAGATGCTTCTCCTCCTAATTTATCATT |
| pOB-ezrA-up-R | ATTCGAGCTCGGTACCCGGGTTTTAAATTAATAAAAAAAACACCCACAATT |
| pOB-ezrA-down-F | CACTATAGAATACTCAAGCTTACTCCTTAATTTCCTCATAAATGATGA |
| pOB-ezrA-down-R | GGATCAACTTTGGGAGAGAGAAACTAGTATGTAGTTATACTTAAATAATATGAGC |
| pOB-TetR-F | TAAATTAGGAGGAGAAGCATCTGCAGATAGTGTACGTAAAAAGA |
| pOB-TetR-R | GTATAACTACATACTAGTTTCTCTCTCCCAAAGTTGATCCC |
| ftsZ-eyfp-F | ACATGGCCATGTCAGGTTCAG |
| ftsZ-eyfp-R | GGCGCGCCTTATTTATATAATTC |
| FGFtsZXhoI-F | CTCGAGATGTTAGAATTTGAACAAGG |
| FGFtsZEcoRI-R | TTAGAATTCACGTCTTGTTCTTCTTGAA |
| FGFtsZNheI-F | GTTGCTAGCATGTTAGAATTTGAACAAGG |
| FGFtsZAscI-R | GTTGGCGCGCCTTATCCCAGACCCGGTTTAC |

DOI: https://doi.org/10.7554/eLife.32057.034

