## [Decision Letter]

Thank you for submitting your article "Molecular coordination of *Staphylococcus aureus* cell division" for consideration by *eLife*. Your article has been reviewed by two peer reviewers, and the evaluation has been overseen by a Reviewing Editor and Gisela Storz as the Senior Editor. The reviewers have opted to remain anonymous.

The reviewers have discussed the reviews with one another and the Reviewing Editor has drafted this decision to frame the key concerns we feel must be addressed before a binding decision may be made on your submission.

Overall, the referees are in agreement that the work potentially represents new insight into cell wall synthesis and division in *S. aureus*. However, both reviewers raised serious concerns regarding the experimental methods, quantification, and interpretations, and consequently the major claims of the work. There were also concerns regarding the necessity of some experiments. Before a decision can be made regarding the publication, we ask you to submit a revised application addressing the following comments:

Major comments:

1) Demonstrate the ability of ADA probes to label new septal cell wall synthesis.

The authors are assuming that FDAA incorporation equals PG synthesis, when it is well known that this is not the case, most especially for the ADAs, which are incorporated into the cell wall by transpeptidase reactions. The authors would be correct in stating that crosslinking occurs everywhere in the cell and throughout the septa, but this is not synthesis. Importantly it has been published by two different groups that PBP4 (a LMW transpeptidase) is responsible for the extra septal FDAA staining signal (Monteiro, 2015 doi:10.1038/ncomms9055), (Gautam, 2015 DOI: 10.1021/jacs.5b02972). These two papers reveal that a large amount of FDAA incorporation occurs via PBP4, but this is due to crosslinking of FDAA's into the existing material and does not reflect the insertion of new material.

Thus, if the authors want to claim that A) PG synthesis occurs throughout the septa, and B) there are no hotspots (or puncta) of PG synthesis in the ring, they need to Redo their FDAA labeling and STORM in PBP4 null cells at different pulse lengths, so that they can (as much as possible) remove this outside "noisy" signal, and truly focus on where synthesis occurs.

To prove this further, they should also conduct various ranges of pulses of ADA-DA (e.g., 5s, 15s, 30s, 1 min) as these are believed to be reflective of synthesis, and not simply crosslinking. These should be done in PBP4 null cells as well.

Finally, the authors went to great detail to make the HADA probe and then do not use it for any significant experiment. They also showed no evidence of incorporation of this probe into the PG (Figure 3—figure supplement 1 only has data for the ADA probe). The ADA-DA probe could be used throughout the manuscript – thereby allowing results to be accurately compared without having to worry about artifacts of the PG-probe. The HADA probe was not used in any significant experiment and could be removed without altering the conclusions of the manuscript.

2) Provide quantitative measurements specifying the localization precision and spatial resolution of PALM/STORM images.

The authors must determine (or at least estimate their precision of localization) in both XY and Z, for eYFP and Alexa 647. Many of the arguments made in Figure 1 and Figure 3 do indeed hinge on their precision of localization. They dismiss the need for this, stating – "It is challenging to determine resolution in localisation microscopy images. However, it is more important to determine which image features observed are representative of biological structures, then to determine absolute resolution." This reviewer adamantly disagrees with this statement, as the conclusions drawn from the widths of their distributions in Figure 1 and Figure 3 rely on their precision of localization, and thus far it is not clear if they have achieved adequate precision to make claims about the widths of these distributions.

To overcome this issue, the authors have attempted to fit their data in Figure 1 with different precisions (up to 60 nm). However, several later statements made in the paper suggest that their precision may indeed not even be that accurate: First, they state their prevision for localization of 647 is <70nm. This is a troubling measure for multiple reasons, as eYFP is significantly dimmer than 647. This makes this reviewer fear they may not have enough photons even to approach 70nm, close to the fitting they used in Figure 1 to justify these measures. Second, while it appears they used flat septa in Figure 1 (in the XY plane), in later figures they used 3D-Storm, which then raises the issue that there is a much less precision in Z relative to XY. This is not raised nor acknowledged in the paper, and how it affects the data is not clear. For example, in Figure 1 and Figure 3, what angle were these septa in that they analyzed? (This complicates their data). Were they flat in XY, which would give the best resolution or were they angled up in Z, or tilted in some combination of XYZ? This is a key issue, as the spread, noise, and asymmetry in these distributions may arise from their precisions of each different septa, and where it is in the Z plane.

So, overall, the authors must report their localization precision in XY, and Z, for both eYFP as well as 647. This is best done (amount other ways) with pure protein immobilized on coverslips, as well known. This should allow them to accurately determine the height of the peak of a single molecule and FWHM of each localization. On an NSTORM system, the precision in Z can also be easily determined, as these systems have piezo stages.

Alternatively, at the very least the authors should at minimally report the computed (localization) precision from frame by frame peak heights in their data, as detailed in doi:10.1038/nmeth.1447. While this does not account for other sources of noise (such as instrument vibrations), it will at least give an estimate. If their analyses are conducted on complexity flat rings in the XY plane, knowing the precision can help the authors gauge the accuracy of their claims. However, if they are drawing on data that involves rings that traverse the Z direction, they should do determine, or at least give a rough calculation of the loss of precision caused by the Z-direction, and use that "worse" precision estimate to determine if they can make the claims in the paper.

3) Re-examine the claim that FtsZ and EzrA are not localized in a thin ring around the division plane, but that these proteins are in much wider bands.

There are a few problems with their analysis. First, the method by which they analyzed these distributions appears to be erroneous. They stated that they looked at the localizations at distances from centers of the circles. While this would make sense if the septa were indeed perfect circles (as shown in their simulated data), but examining the examples they show it appears most of many of these septa are not perfect circles, rather they are extended ellipses. Thus, measuring the distance from the center is not a valid measure of thickness (or distance to the edge), and thus this assumption is likely distorting their conclusions. If they want to back up this claim, it would be better they fit an ellipse to the density each septal ring, and then plotted the distribution of localizations around that line around the septa.

Second, as described above, to make the claim that FtsZ and EzrA are not in a thin "ring" they must determine their localization precision (discussed more in 2 above). Currently, they attempt to get around this by estimating the spread with different localization precisions. This is not adequate, as they have not even demonstrated they have 60nm precision (the largest value they use).

Other concerns:

1) In Figure 4—figure supplement 1, septal peptidoglycan synthesis actually looks well-localized to my eyes for 15-20 minutes and even to some extent after 30-35 minutes of PC190723 treatment, well beyond the time when FtsZ and EzrA have delocalized. Do the authors have an explanation for why this might be? It seems contrary to their claim in the manuscript that gross PG synthesis follows FtsZ and EzrA localization, although this is certainly true at later time points.

2) The Slimfield analysis, while nice, adds nothing to this paper or story. All this gives them is the diffusion coefficient of EzrA. First, how is this measure pertinent to the story? That is not made clear. Second, there also appears to be an immobile fraction, but the authors report the characteristics of the diffusive form. Is this the biologically relevant form? If EzrA is treadmilling with FtsZ, it should be immobile, and the diffusive molecules not associated with FtsZ.

Furthermore, the authors should rethink their statement about EzrA moving around the cell with FtsZ – that would not be expected for molecules associate with treadmilling filaments.

3) Figure 3—figure supplement 2 and C should present the same brightness and contrast range for comparison.

4) In the last paragraph of the Discussion, it is difficult to follow how the data presented support or do not support a model of "machines moving through the cell depositing peptidoglycan." The article argues persuasively that septal PG synthesis is not confined to the inner edge of the invaginating septum in Staph, but doesn't include much detail about the molecular mechanism of insertion (which is fine, and not necessary for this article). Given that FtsZ and EzrA are colocalized with PG incorporation in this system as well, it is a question of how the authors rule out a model where these hypothetical molecular machines do exist, but are distributed across the entire septum rather than localized in one region- such a system seems consistent with the results presented here. It is also confusing that these proteins are not always physically adjacent- it seems like most of the article has been arguing the contrary, that a wider FtsZ/EzrA localization band gives rise to more broadly distributed septal PG insertion.

5) The use of the Alexa Fluor 647 NHS ester should be clarified. This reagent non-specifically labels proteins in the cell wall.

6) As unnatural amino acids are used, could the authors comment on how they think these additions could change the structure of the cell wall? Could the probes possibly effect the placement of the machines?

7) Could the authors explain what "gross peptidoglycan synthesis" is? How does this play into their model?

8) The treatment with labeled vancomycin is not standard – usually unlabeled vancomycin is added with the labeled vancomycin. It is not certain if vancomycin is capable of labeling a D-Ala-D-Ala terminus that has been labeled with a fluorophore – this is very close to the binding pocket. That could be a reason why they do not see labeling there? What do these experiments add to their conclusions?

[Editors' note: further revisions were requested prior to acceptance, as described below.]

Thank you for resubmitting your work entitled "Molecular coordination of *Staphylococcus aureus* cell division" for further consideration at *eLife*. Your revised article has been favorably evaluated by Gisela Storz (Senior Editor) and a Reviewing Editor.

The manuscript has been improved but there are some remaining issues that need to be addressed before acceptance, as outlined below:

1) Calculation of the spatial resolution of EzrA and FtsZ imaging:

Localization precision only tells how accurately one can determine the position of one localization. It is usually much better than the actual spatial resolution one can achieve, which is compounded by the Nyquist resolution (calculated by labeling density), and experimental resolution (calculated by the spread of repeat localization). Of these two, the experimental resolution often is the limiting factor, and should be reported. The authors cited reasons for not doing the latter calculation coltharp, but this measurement can be done by using the fixed bacteria instead of purified, in-vitro samples. See Endesfelder et al., 2014, Churchman LS et al., 2006, Biophysical J, 90(2):668-671, and Coltharp et al., 2012.

2) PBP4 null experiment:

There was one image (Figure 3—figure supplement 3) qualitatively suggesting that 15s ADA-DA and ADA incorporation in the absence of PBP4 is homogeneous. Please provide enough statistics to support this conclusion, i.e., number of cells, autocorrelation function and labeling density.

---

## [Author Response]

Overall, the referees are in agreement that the work potentially represents new insight into cell wall synthesis and division in S. aureus. However, both reviewers raised serious concerns regarding the experimental methods, quantification, and interpretations, and consequently the major claims of the work. There were also concerns regarding the necessity of some experiments. Before a decision can be made regarding the publication, we ask you to submit a revised application addressing the following comments:Major comments:1) Demonstrate the ability of ADA probes to label new septal cell wall synthesis.The authors are assuming that FDAA incorporation equals PG synthesis, when it is well known that this is not the case, most especially for the ADAs, which are incorporated into the cell wall by transpeptidase reactions. The authors would be correct in stating that crosslinking occurs everywhere in the cell and throughout the septa, but this is not synthesis. Importantly it has been published by two different groups that PBP4 (a LMW transpeptidase) is responsible for the extra septal FDAA staining signal (Monteiro, 2015 doi:10.1038/ncomms9055), (Gautam, 2015 DOI: 10.1021/jacs.5b02972). These two papers reveal that a large amount of FDAA incorporation occurs via PBP4, but this is due to crosslinking of FDAA's into the existing material and does not reflect the insertion of new material.Thus, if the authors want to claim that A) PG synthesis occurs throughout the septa, and B) there are no hotspots (or puncta) of PG synthesis in the ring, they need to Redo their FDAA labeling and STORM in PBP4 null cells at different pulse lengths, so that they can (as much as possible) remove this outside "noisy" signal, and truly focus on where synthesis occurs.To prove this further, they should also conduct various ranges of pulses of ADA-DA (e.g., 5s, 15s, 30s, 1 min) as these are believed to be reflective of synthesis, and not simply crosslinking. These should be done in PBP4 null cells as well.Finally, the authors went to great detail to make the HADA probe and then do not use it for any significant experiment. They also showed no evidence of incorporation of this probe into the PG (Figure 3—figure supplement 1 only has data for the ADA probe). The ADA-DA probe could be used throughout the manuscript – thereby allowing results to be accurately compared without having to worry about artifacts of the PG-probe. The HADA probe was not used in any significant experiment and could be removed without altering the conclusions of the manuscript.

We agree that PBP4 activity could have an effect on the observed FDAA incorporation patterns. Thus, a *pbp4* mutant was used. Strain SH4425 (SH1000 *pbp4*) has no growth defect compared to its parent and shows identical, overall peptidoglycan synthesis as measured by ^14^C GlcNAc incorporation (Figure 3—figure supplement 3). Using both HADA and ADA-DA labelling for 5 min, SH4425 shows decreased incorporation of either probe compared to SH1000. However, this reduction is across the entire cell and not as a result of loss of off septal labelling per se. This is contrary to published data (1) that can be explained by low level labelling in the previous study. To further map synthesis in SH4425 (*pbp4*), STORM was carried out using cells labelled with ADA and ADA-DA for 5min and 15s (Figure 3—figure supplement 3). This clearly demonstrates that there is labelling across the septum, around the cell periphery and not limited to apparent foci of insertion. This is corroborated by STORM analysis of SH1000 labelled for 15s with ADA-DA (Figure 3—figure supplement 2).

The HADA probe has been used extensively in previous FDAA studies (2, 3) and now here in the further analysis of SH4425 (*pbp4*) (Figure 3—figure supplement 3).

2) Provide quantitative measurements specifying the localization precision and spatial resolution of PALM/STORM images.The authors must determine (or at least estimate their precision of localization) in both XY and Z, for eYFP and Alexa 647. Many of the arguments made in Figure 1 and Figure 3 do indeed hinge on their precision of localization. They dismiss the need for this, stating – "It is challenging to determine resolution in localisation microscopy images. However, it is more important to determine which image features observed are representative of biological structures, then to determine absolute resolution." This reviewer adamantly disagrees with this statement, as the conclusions drawn from the widths of their distributions in Figure 1 and Figure 3 rely on their precision of localization, and thus far it is not clear if they have achieved adequate precision to make claims about the widths of these distributions.To overcome this issue, the authors have attempted to fit their data in Figure 1 with different precisions (up to 60 nm). However, several later statements made in the paper suggest that their precision may indeed not even be that accurate: First, they state their prevision for localization of 647 is <70nm. This is a troubling measure for multiple reasons, as eYFP is significantly dimmer than 647. This makes this reviewer fear they may not have enough photons even to approach 70nm, close to the fitting they used in Figure 1 to justify these measures. Second, while it appears they used flat septa in Figure 1 (in the XY plane), in later figures they used 3D-Storm, which then raises the issue that there is a much less precision in Z relative to XY. This is not raised nor acknowledged in the paper, and how it affects the data is not clear. For example, in Figure 1 and Figure 3, what angle were these septa in that they analyzed? (This complicates their data). Were they flat in XY, which would give the best resolution or were they angled up in Z, or tilted in some combination of XYZ? This is a key issue, as the spread, noise, and asymmetry in these distributions may arise from their precisions of each different septa, and where it is in the Z plane.So, overall, the authors must report their localization precision in XY, and Z, for both eYFP as well as 647. This is best done (amount other ways) with pure protein immobilized on coverslips, as well known. This should allow them to accurately determine the height of the peak of a single molecule and FWHM of each localization. On an NSTORM system, the precision in Z can also be easily determined, as these systems have piezo stages.Alternatively, at the very least the authors should at minimally report the computed (localization) precision from frame by frame peak heights in their data, as detailed in doi:10.1038/nmeth.1447. While this does not account for other sources of noise (such as instrument vibrations), it will at least give an estimate.

Computed localisation precisions can be obtained using the widely adopted “Thompson” Equation (4) which is the basis of localisation precision determination in both the N-STORM software (used for our 3D imaging) and in ThunderSTORM (used for our 2D imaging). We reproduce this here for reference:⟨(∆x)2⟩=s2+a2/12N+8πs4b2a2N2

This equation has attracted criticism for potentially underestimating localisation precision and a modified version of the equation is presented in the paper referenced by the reviewers (5).

We can use these to summarise calculated localisation precision for example datasets. We fully acknowledge that this information would have been very useful to those reviewing the manuscript and have included it in our amended version (subsection “Distribution of divisome components during septation”, fourth paragraph and subsection “Peptidoglycan synthesis in *S. aureus* does not occur in discrete foci”, second paragraph). Determining localisation precision by repeatedly imaging immobilised fluorophores would probably lead to an underestimate of localisation precision as background will be higher in our bacteria than on a clean surface (where there would also be near field coupling to the glass, boosting the detection efficiency) – we have therefore not done this.

FluorophoreLocalisation Precision (nm) (Quoted as a standard deviation)MethodAlexa Fluor 647 (3D with cylindrical lens)9.8 (s.d. 3.5)N-STORM Software (Thompson)YFP24 (s.d. 8.5)ThunderSTORM (Thompson)27 (s.d. 8.7)Matlab Function (Mortensen’s modified Thompson)

Substantial variability over individual blinks is to be expected (due to variability in photon counts/blink). Although experience tells us that YFP is dimmer than Alexa Fluor 647 for confocal and epifluorescence imaging, we have found than an improved buffer results in an acceptable number of photons per blink for single molecule imaging (as have others (6)). We used D_2_O to make the buffer to boost photon counts (7). Alexa Fluor 647 still performed much better than YFP.

The critical point here is that the features we see in our images, on which we base our conclusions, are on a much larger scale than our localisation precision. Our approach of comparing simulations and microscope data, and exploring pessimistic scenarios regarding performance, was rooted in concerns that good localisation precision alone does not always equate to reliable data.

If their analyses are conducted on complexity flat rings in the XY plane, knowing the precision can help the authors gauge the accuracy of their claims. However, if they are drawing on data that involves rings that traverse the Z direction, they should do determine, or at least give a rough calculation of the loss of precision caused by the Z-direction, and use that "worse" precision estimate to determine if they can make the claims in the paper.

We have adapted our methods to use ellipse fitting as an alternative to circle fitting (see our response to point 3 below, subsection “Distribution of divisome components during septation”, fourth paragraph and Figure 1—figure supplement 3). If the elliptical shape of the rings is the result of the effective projection of a circular ring onto the imaging plane, we can calculate how much the circle would have to be tilted and thus the range of distances perpendicular to the imaging plane in which we would expect to find molecules.

In localisation microscopy, blinks a long way from the “absolute” plane of focus are more poorly localised than those “in focus”. A justified concern is that this would mean that for some molecules localisation precision would be very poor, and falsely lead us to our conclusion that the molecules are not in a very thin ring. However, the spread of localisations in z is within the range for which we expect our objective to deliver good data (see (8) for a typical z range). Furthermore, we have now provided calculated localisation precisions which take into account potential reduced precision (due to fewer photons or a broader Gaussian). We demonstrate in our response to comment 3, below, that our localisation precision is good enough to support our conclusions.

3) Re-examine the claim that FtsZ and Ezra are not localized in a thin ring around the division plane, but that these proteins are in much wider bands.There are a few problems with their analysis. First, the method by which they analyzed these distributions appears to be erroneous. They stated that they looked at the localizations at distances from centers of the circles. While this would make sense if the septa were indeed perfect circles (as shown in their simulated data), but examining the examples they show it appears most of many of these septa are not perfect circles, rather they are extended ellipses. Thus, measuring the distance from the center is not a valid measure of thickness (or distance to the edge), and thus this assumption is likely distorting their conclusions. If they want to back up this claim, it would be better they fit an ellipse to the density each septal ring, and then plotted the distribution of localizations around that line around the septa.

Carrying out this analysis based on an ellipse rather than a circle complicates it as the circle fit was a primarily means of shifting to a polar co-ordinate system, not to analyse the distribution of molecules relative to a circle of a specific radius – the analysis is independent of the fitted radius. However, we have adapted the analysis method to a fitted ellipse to test whether the approach proposed by the reviewers is advantageous or impacts on our conclusions.

We used an elliptical ring divided into blocks to inform us of the way in which the number of localisations changes as we go around the ring. In the illustration below we have used different shades of blue to show which blocks contain more or less molecules. Yellow localisations are included in the analysis, whilst grey ones are not.

**Author response image 2. respfig2:** 

Second, as described above, to make the claim that FtsZ and Ezra are not in a thin "ring" they must determine their localization precision (discussed more in 2 above).

Please see our response to point 2 regarding localisation precision.

Currently, they attempt to get around this by estimating the spread with different localization precisions. This is not adequate, as they have not even demonstrated they have 60nm precision (the largest value they use).

We determined how close each molecule was to the closest point on the fitted ellipse and compared this distribution of distances to a simulation (applying localisation precision from a normal distribution with a mean of 27 nm and a standard deviation of 8.7 nm). We again concluded that the variation in molecule position cannot be accounted for by localisation error alone, and that the molecules were not in a very thin ring, even if this ring is an ellipse (Figure 1—figure supplement 3).

Other concerns:1) In Figure 4—figure supplement 1, septal peptidoglycan synthesis actually looks well-localized to my eyes for 15-20 minutes and even to some extent after 30-35 minutes of PC190723 treatment, well beyond the time when FtsZ and EzrA have delocalized. Do the authors have an explanation for why this might be? It seems contrary to their claim in the manuscript that gross PG synthesis follows FtsZ and EzrA localization, although this is certainly true at later time points.

We agree that the image of HADA cells at 15-20 minutes looks more similar to that at 0-5 minutes than do the images of FtsZ-SNAP or EzrA-YFP. However, there are at 15-20 minutes some cells within the field shown that have clear defects in peptidoglycan insertion pattern, showing asymmetric or punctate insertion. Figure 4—figure supplement has been amended to highlight these features. In all cases the images for the three channels are similar enough to support the conclusion that EzrA, FtsZ and peptidoglycan insertion are all in the same places.

2) The Slimfield analysis, while nice, adds nothing to this paper or story. All this gives them is the diffusion coefficient of EzrA. First, how is this measure pertinent to the story? That is not made clear. Second, there also appears to be an immobile fraction, but the authors report the characteristics of the diffusive form. Is this the biologically relevant form? If EzrA is treadmilling with FtsZ, it should be immobile, and the diffusive molecules not associated with FtsZ.Furthermore, they authors should rethink their statement about Ezra moving around the cell with FtsZ – that would not be expected for molecules associate with treadmilling filaments.

The Slimfield microscopy and associated analysis enable precise quantitation of molecular mobility over a rapid millisecond time scale comparable to molecular diffusion rates inside live cells i.e. it allows us to quantify mobile and immobile populations of EzrA molecules directly inside live cells. However, it also enables direct quantitation of the EzrA copy number on a cell-by-cell basis – this is achieved through a process of determining the characteristic brightness of single YFP molecules inside each cell from Slimfield images and then deconvolving these images directly. Slimfield gives us an unprecedented insight into precisely how much EzrA is present in live cells and how its molecular mobility changes depending on whether or not it is incorporated into a septum structure; this level of robust quantification indicates a more complex behavior than simple treadmilling alone, and we believe it has value in supporting a more heterogeneous model for the cell growth and division processes than has been suggested previously, when viewed in the context of the other data presented in our paper.

Specifically, the full distribution of measured microscopic diffusion coefficients (D) for all tracked EzrA-YFP foci are given in the plots of Figure 1—figure supplement 4; these indicate a mixture of both a relatively immobile and two relatively mobile populations. The characteristic D values reported in the main text are overall mean values which are calculated over the full range of measured D, either inside or outside the septum, and so include contributions from both the immobile and two mobile components. We have now reworded this section to clarify this specifically to avoid confusion (subsection “Distribution of divisome components during septation”, last paragraph). The text in this section has also been amended to indicate clearly that copy number was determined directly by Slimfield.

Our measured tracking localization precision of a few tens of nm (9) indicates that rates of diffusion values below ~0.02 µm^2^/s will appear predominantly in the apparent immobile component of the distribution of D values, even if they have a real underlying mobility. The mean speed of putative FtsZ treadmilling estimated from *B. subtilis* (3) is only ~30nm/s, which we estimate would be sufficiently slow to appear predominantly in the immobile component over the typical time scales of our Slimfield tracking experiments here, and so we agree with the reviewer that any putative treadmilling of EzrA at this equivalent mean speed, if present in *S. aureus*, would appear in this apparent immobile fraction. Conversely, however, this measurement for the proportion of the apparent immobile fraction sets an upper bound for the proportion of EzrA which may be treadmilling in association with FtsZ. EzrA present in the most mobile fraction is unlikely to be associated with FstZ, we agree with the reviewer. However, in the 3-γ function mobility model, which fits the observed distribution of D values well, the intermediate mobility fraction has been interpreted previously in other cellular systems as indicating transient dynamic interactions(10), so we cannot entirely exclude the possibility that this may be due to transient association of EzrA with FtsZ i.e. that it is also the ‘biologically relevant form’ as that present in the apparent immobile fraction. However, we can then derive an important result that at least ~1/3 of EzrA is not treadmilling with FtsZ. In other words, we cannot account for the observed mobility of EzrA by a simple treadmilling model alone in which all EzrA is tightly associated to FtsZ, but rather the real cellular behavior is more nuanced than this. We have now amended the text to include discussion of this (see the aforementioned subsection).

3) Figure 3—figure supplement 2 and C should present the same brightness and contrast range for comparison.

Different FDAAs have different brightness in the microscope, therefore using numerically the same brightness and contrast settings will not be optimal for all fluorophores. Images were re-processed to be optimal for individual channels where the background was not black and the highest pixel values were not saturated, therefore no details will be hidden. Since STORM data is a plot of localisations rather than an image the same contrast cannot be applied. We hope that the revised figure will make it easier for readers to compare the images (Figure 3—figure supplement 2).

4) In the last paragraph of the Discussion, it is difficult to follow how the data presented support or do not support a model of "machines moving through the cell depositing peptidoglycan." The article argues persuasively that septal PG synthesis is not confined to the inner edge of the invaginating septum in Staph, but doesn't include much detail about the molecular mechanism of insertion (which is fine, and not necessary for this article). Given that FtsZ and EzrA are colocalized with PG incorporation in this system as well, it is a question of how the authors rule out a model where these hypothetical molecular machines do exist, but are distributed across the entire septum rather than localized in one region- such a system seems consistent with the results presented here. It is also confusing that these proteins are not always physically adjacent- it seems like most of the article has been arguing the contrary, that a wider FtsZ/EzrA localization band gives rise to more broadly distributed septal PG insertion.

We have revised the last paragraph of the Discussion to incorporate the reviewers points so as not to over-interpret our data. We agree that our data does not rule our molecular machines, but that it provokes questions as to the nature, stability and dynamism of these conceptual groupings of proteins.

5) The use of the Alexa Fluor 647 NHS ester should be clarified. This reagent non-specifically labels proteins in the cell wall.

The NHS ester labels amine groups which are present in both peptidoglycan and proteins. The reagent was used to indicate the location of the entirety of the cell wall relative to other cell components – we make no claim that it is a peptidoglycan specific label. This has been clarified in the text (subsection “Distribution of divisome components during septation”, seventh paragraph).

6) As unnatural amino acids are used, could the authors comment on how they think these additions could change the structure of the cell wall? Could the probes possibly effect the placement of the machines?

We have combined unnatural amino acid labelling with imaging of EzrA-YFP and FtsZ-YFP using diffraction limited microscopy and see no difference in the distribution of the proteins. The reviewer is right to point out that localised alterations to peptidoglycan chemistry may affect the localisation of proteins involved with growth and division, however if this occurs, it does so at scales we cannot currently access. We assume the number of peptides altered by our amino acid (or dipeptide) labelling is a very small proportion of the total.

**Author response image 3. respfig3:** Effect of HADA labelling on EzrA-eYFP and FtsZ-SNAP TMR-Star localisation. *S. aureus* strain producing EzrA-eYFP grown without (**a**) and with HADA for 5 min (**b**). SNAP-Cell TMR-Starlabelled *S. aureus* strain producing FtsZ-SNAP grow without (**c**) and with HADA for 5 min (**d**). Images are average intensity fluorescence projections of z stacks. Scale bars 5 𝝻m.

7) Could the authors explain what "gross peptidoglycan synthesis" is? How does this play into their model?

This was not a great choice of words – we have amended the text for clarity (throughout).

8) The treatment with labeled vancomycin is not standard – usually unlabeled vancomycin is added with the labeled vancomycin.

This is only the case if fluorescent vancomycin is purchased ready labelled. For “home-made” fluorescent vancomycin, no unlabelled drug needs to be added (11).

It is not certain if vancomycin is capable of labeling a D-Ala-D-Ala terminus that has been labeled with a fluorophore – this is very close to the binding pocket. That could be a reason why they do not see labeling there?

It is likely that vancomycin cannot bind side chains that are not D-ala-D-ala terminated – incorporation of D-serine (for example) instead of D-alanine in the positions blocks vancomycin binding. However, many D-ala-D-ala terminated peptides remain in ADA labelled bacteria, resulting in a strong vancomycin fluorescence signal over the entire cell surface.

What do these experiments add to their conclusions?

Combining vancomycin and ADA labelling reveals regions where there is vancomycin binding, but no ADA incorporation. This confirms that the regions within partially formed septa that are not labelled with ADA, are in fact filled with peptidoglycan, to which the vancomycin can bind (subsection “Peptidoglycan synthesis in *S. aureus* does not occur in discrete foci”, third paragraph).

References

1) Monteiro JM, Fernandes PB, Vaz F, Pereira AR, Tavares AC, Ferreira MT, et al. Cell shape dynamics during the staphylococcal cell cycle. Nature Communications. 2015;6:8055.

2) Kuru E, Hughes HV, Brown PJ, Hall E, Tekkam S, Cava F, et al. *In Situ* probing of newly synthesized peptidoglycan in live bacteria with fluorescent ᴅ-amino acids. Angewandte Chemie Int Ed. 2012;51(50):12519-23.

3) Bisson Filho AW, Hsu YP, Squyres GR, Kuru E, Wu F, Jukes C, et al. Treadmilling by FtsZ filaments drives peptidoglycan synthesis and bacterial cell division. Science. 2017;355:739-43.

4) Thompson RE, Larson DR, Webb WW. Precise Nanometer Localization Analysis for Individual Fluorescent Probes. Biophysical Journal. 2002;82(5):2775-83.

5) Mortensen KI, Churchman LS, Spudich JA, Flyvbjerg H. Optimized localization analysis for single-molecule tracking and super-resolution microscopy. Nature Methods. 2010;7:377.

6) Jusuk I, Vietz C, Raab M, Dammeyer T, Tinnefeld P. Super-Resolution Imaging Conditions for enhanced Yellow Fluorescent Protein (eYFP) Demonstrated on DNA Origami Nanorulers. Scientific Reports. 2015;5:14075.

7) Ong WQ, Citron YR, Schnitzbauer J, Kamiyama D, Huang B. Heavy water: a simple solution to increasing the brightness of fluorescent proteins in super-resolution imaging. Chemical Communications. 2015;51(70):13451-3.

8) Palayret M, Armes H, Basu S, Watson AT, Herbert A, Lando D, et al. Virtual-'Light-Sheet' Single-Molecule Localisation Microscopy Enables Quantitative Optical Sectioning for Super-Resolution Imaging. PLOS ONE. 2015;10(4):e0125438.

9) Wollman AJM, Leake MC. Millisecond single-molecule localization microscopy combined with convolution analysis and automated image segmentation to determine protein concentrations in complexly structured, functional cells, one cell at a time. Faraday Discussions. 2015;184(0):401-24.

10) Stracy M, Lesterlin C, Garza de Leon F, Uphoff S, Zawadzki P, Kapanidis AN. Live-cell superresolution microscopy reveals the organization of RNA polymerase in the bacterial nucleoid. Proceedings of the National Academy of Sciences. 2015;112(32):E4390-E9.

11) Daniel RA, Errington J. Control of Cell Morphogenesis in Bacteria: Two Distinct Ways to Make a Rod-Shaped Cell. Cell. 2003;113:767-76.

[Editors' note: further revisions were requested prior to acceptance, as described below.]

The manuscript has been improved but there are some remaining issues that need to be addressed before acceptance, as outlined below:1) Calculation of the spatial resolution of EzrA and FtsZ imaging:Localization precision only tells how accurately one can determine the position of one localization. It is usually much better than the actual spatial resolution one can achieve, which is compounded by the Nyquist resolution (calculated by labeling density), and experimental resolution (calculated by the spread of repeat localization). Of these two, the experimental resolution often is the limiting factor, and should be reported. The authors cited reasons for not doing the latter calculation coltharp, but this measurement can be done by using the fixed bacteria instead of purified, in-vitro samples. See Endesfelder et al., 2014, Churchman LS et al., 2006, Biophysical J, 90(2):668-671, and Coltharp et al., 2012.

We fully acknowledge the importance of understanding localisation precision and resolution in this type of study. We have used the NeNA method (executed in the Lama software package) to extract mean values from our experimental data on fixed cells. These are 7.5 nm for Alexa Fluor 647 and 16.23 nm for YFP, in both cases smaller than the “theoretical precision”. (subsection “Distribution of divisome components during septation”, fourth paragraph and subsection “Peptidoglycan synthesis in *S. aureus* does not occur in discrete foci”, second paragraph).

We are cautious about localisation precision estimates and would like to re-emphasise that our analysis methods take localisation precision into account, and we have demonstrated that our conclusions would hold even if we were to have vastly underestimated localisation precision.

2) PBP4 null experiment:There was one image (Figure 3—figure supplement 3) qualitatively suggesting that 15s ADA-DA and ADA incorporation in the absence of PBP4 is homogeneous. Please provide enough statistics to support this conclusion, i.e., number of cells, autocorrelation function and labeling density.

We have applied the elliptical analysis method suggested in the previous review to *S. aureus* labelled for 15 seconds with ADA or ADA-DA, both wild-type (SH1000) and lacking the gene encoding PBP4 (SH1000 *pbp4*). There were 10 cells in each group. We saw no substantial difference in autocorrelation function. This supports the qualitative observation that there is no major difference in incorporation distribution of ADA or ADA-DA, in the presence or absence of PBP4 (subsection “Peptidoglycan synthesis in *S. aureus* does not occur in discrete foci”, second paragraph and Figure 3—figure supplement 3).

**Author response image 4. respfig4:** 

Mean labelling densities (to the nearest one localisation) were:

SampleLabellingDensityWild TypeADA436 localisations per cell*pbp4*ADA881 localisations per cellWild TypeADA-DA683 localisations per cell*pbp4*ADA-DA641 localisations per cell

It is risky to compare numbers of localisations (and thus any kind of density) when using Alexa Fluor 647 (or a similar fluorophore) as each molecule may blink multiple times within the duration an experiment. This comparison is not advisable for our data set as we terminated imaging when no more blinks could be collected, rather than after a fixed time period, and adjusted laser power for maximum blinks.